# SPACE GROUP CONDITIONAL FLOW MATCHING

## ABSTRACT

Inorganic crystals are periodic, highly-symmetric arrangements of atoms in three-dimensional space. Their structures are constrained by the symmetry operations of a crystallographic *space group* and restricted to lie in specific affine subspaces known as *Wyckoff positions*. The frequency an atom appears in the crystal and its rough positioning are determined by its Wyckoff position. Most generative models that predict atomic coordinates overlook these symmetry constraints, leading to unrealistically high populations of proposed crystals exhibiting limited symmetry. We introduce Space Group Conditional Flow Matching, a novel generative framework that samples significantly closer to the target population of highly-symmetric, stable crystals. We achieve this by conditioning the entire generation process on a given space group and set of Wyckoff positions; specifically, we define a conditionally symmetric noise base distribution and a group-conditioned, equivariant, parametric vector field that restricts the motion of atoms to their initial Wyckoff position. Our form of group-conditioned equivariance is achieved using an efficient reformulation of *group averaging* tailored for symmetric crystals. Importantly, it reduces the computational overhead of symmetrization to a negligible level. We achieve state of the art results on de novo generation and ground truth Wyckoff conditioned crystal structure prediction benchmarks.

## 1 INTRODUCTION

Crystals are solid materials characterized by a periodic arrangement of their constituent atoms. The crystalline structure is fundamentally represented by three components: lattice parameters (defining the geometry of the repeating unit cell), fractional coordinates (specifying the position of each atom within the cell), and the identity of the atom at each location. The discovery of novel crystalline structures is critical for material design and recent progress in generative modeling has demonstrated a promising approach to this problem. However, most existing generative methods overlook key crystallographic properties, the space group and Wyckoff positions, making it challenging for them to generate non-trivial symmetric crystals.

A crystal's space group, a subgroup of the Euclidean group $E(n)$, fully describes the symmetry of the atoms arranged within the unit cell. Beyond its correlation with many optical, electrical, magnetic, and structural properties (Chen et al., 2022; Tang et al., 2019; Malgrange et al., 2014; Yang, 2005), the space group imposes constraints on atomic locations and lattice structure. These manifests in form of Wyckoff positions, which are sets of symmetrically equivalent points within a unit cell. More generally, the Wyckoff positions of a space group partition the unit cell according to the structure of the orbits induced by the group (see fig. 2 for a 2D example).

In this work, we develop a generative model that samples crystals conditioned on a given space group and associated Wyckoff positions. This approach offers two key benefits: (1) it provides greater control over the structure and symmetry of the generated crystals, and (2) it can leverage the lower-dimensional constraints imposed by Wyckoff positions for improving generation. In contrast to prior methods that incorporate space group information but rely on projection steps to correct atomic placements (Jiao et al., 2024; Levy et al., 2025), our model is designed to inherently preserve the assignment of atoms to their designated Wyckoff positions throughout the generation process.

Our proposed model, *Space Group Conditional Flow Matching* (SGFM), is based on the *Flow Matching* (FM) generative framework (Albergo & Vanden-Eijnden, 2022; Liu et al., 2022; Lipman et al., 2022). We chose the FM framework for two key reasons: it allows us to use customized source

Figure 1: Visualization of the main components of SGFM. (a) Wyckoff position noise prior. General points are sampled randomly and projected according to the conditioned Wyckoff positions. (b) Space group equivariant vector field. The equivariance of the model combined with the $G$-symmetry of the input crystal ensures that atoms preserve their symmetry structure. (c) Comparison between standard group averaging and our optimized formulation. The $G$-symmetry of the input crystal allows space group operations to be replaced by their corresponding permutations (denoted using $\sigma$). When combined with the $S_n$-equivariance of $u$, this reduces the number of required forward passes per crystal. For visualization purposes, heavy arrows indicate expensive model forward passes.

distributions and provides known conditions for generating data that respects specified symmetries (Köhler et al., 2020). Based on these advantages, we designed SGFM with two main components: A space group and Wyckoff position conditioned noise prior, which have positive support only for crystal structures that adhere to the symmetry constraints described by the Wyckoff positions; A group conditioned equivariant vector field, which is a single neural network architecture that is able to support arbitrary space group equivariance. Equivariance is achieved through Group Averaging (GA) (Yarotsky, 2022), a symmetrization technique that projects arbitrary functions onto their equivariant versions. Although GA is typically computationally expensive and impractical, we introduce an efficient formulation tailored for symmetric crystals, reducing the computational overhead of the symmetrization operator to a negligible level.

The main contributions of this work are as follows:

- We formalized the problem of symmetric crystal generation in terms of distributional symmetry properties (section 3.1), and extended the conditions introduced by Köhler et al. (2020) to enable flow-based models to sample from such distributions (section 3.2).

- We instantiate this flow model as SGFM (section 3.3), which consists of a noise prior conditional on Wyckoff position along with a space group-equivariant vector field, ensuring that the generated crystals preserve the specified symmetries.

- We propose a novel and efficient implementation of GA for symmetric crystals, equivalent in output to the standard GA but significantly more efficient (see fig. 1 (c)), practically minimizing the computational burden of symmetrization for symmetric crystals.

- SGFM achieves state-of-the-art performance on de novo generation (DNG). Compared with a baseline that also receives the ground truth Wyckoff positions, SGFM achieves SOTA on a crystal structure prediction (CSP) benchmark in which atom types are given.

## 2 PRELIMINARIES

**Equivariance & Invariance.** A function $\varphi : \mathcal{X} \to \mathcal{Y}$ is equivariant with respect to a group $G$ if the action of any group element on the input corresponds to a consistent transformation of the output. Equivariance implies $\varphi(g \cdot x) = g \cdot \varphi(x)$ for all $x \in \mathcal{X}$ and $g \in G$. Invariance is a simplified case of equivariance, with all $g \in G$ mapping to a trivial group action on the output space $\varphi(g \cdot x) = \varphi(x)$. Invariance and equivariance also extend to group products: Let $(g_1, g_2) \in G_1 \times G_2$, $\varphi$ is $G_1 \times G_2$ equivariant if $\varphi((g_1, g_2) \cdot x) = (g_1, g_2) \cdot \varphi(x)$. Additionally, *G-invariant distributions* refer to distributions which have an $G$-invariant density function. We will use this to construct SGFM.

**Crystal Representation.** A crystal can be represented using the tuple $c' = (\boldsymbol{L}, \boldsymbol{F}, \boldsymbol{A}) \in \mathcal{C}'$, where $\boldsymbol{L} \in \mathbb{R}^{3 \times 3}$ is a positive-determinant, invertible matrix defining the lengths, angles, and orientation of the positive-volume unit cell; $\boldsymbol{F} \in [0, 1)^{n \times 3}$ denotes the fractional coordinates of $n$ atoms within the unit cell; and $\boldsymbol{A} \in \{0, 1\}^{n \times h}$ is a one-hot matrix indicating the atom types in $\boldsymbol{F}$ from a set of $h$

atom types. We adopt the space group–conditioned lattice parameterization proposed by Jiao et al. (2024), which replaces $L$ with a rotation-invariant vector $k \in \mathbb{R}^6$ of coefficients of a symmetric matrix basis, and represent a crystal as $c = (k, F, A) \in \mathcal{C}$. Further details provided in appendix C.

**Symmetries & Crystals.** Our method focuses on how symmetry groups act on crystals. The permutation group $S_n$ acts on $c$ by permuting the rows of $F$ and $A$. Namely, if $\sigma \in S_n$ is represented by a permutation matrix $P \in \{0, 1\}^{n \times n}$ then $\sigma \cdot c = (k, PF, PA)$. The group of isometries of $\mathbb{R}^3$ known as the Euclidean group $E(3)$, acts on $c$ by applying an orthogonal transformation $R \in O(3)$ and a translation $\tau \in \mathbb{R}^3$ to the fractional coordinates. For a group element $g = (R, \tau) \in E(3)$, the action is defined as $g \cdot c = (k, FR^T + 1_n \tau^T - \lfloor FR^T + 1_n \tau^T \rfloor, A)$ where $1_n \in \{1\}^n$ is a column vector of ones and $\lfloor \cdot \rfloor$ is the element-wise floor function. We further specify the action of the product group on $c$. Let $(g, \sigma) \in G \times S_n$, we define $(g, \sigma) \cdot c := g \cdot (\sigma \cdot c)$. Puny et al. (2021) showed that if $G \leqslant E(3)$ then $g \cdot (\sigma \cdot c) = \sigma \cdot (g \cdot c)$, i.e., the operators commute.

**Space Groups.** The space group concept formalizes the intrinsic symmetry of a crystal. If $c \in \mathcal{C}$ is a crystal and $G \leqslant E(3)$ is its symmetry space group, then for any $g \in G$ there exist a permutation $\sigma \in S_n$ that satisfies the relation $g \cdot c = \sigma \cdot c$, a property we will denote as *G-symmetry*. In essence, any action of the space group is equivalent to a permutation of the atom positions, fig. 2 visualizes this property using a 2D example. The p4mm symmetry group includes rotations by angles of $\frac{\pi z}{2}$ for $z \in \mathbb{Z}$. The corresponding permutation is invisible, without fabricated labels, because it rearranges the positions of identical shapes. Two Crystals $c_1, c_2 \in \mathcal{C}$ are *Mutually G-Symmetric* if every space group element $g$ corresponds to the same permutation $\sigma$ on both crystals. Formally, $c_1$ and $c_2$ are mutually $G$-symmetric if $g \cdot c_1 = \sigma \cdot c_1 \iff g \cdot c_2 = \sigma \cdot c_2$. There exist 230 distinct space groups in three-dimensional crystallography. Owing to the intrinsic periodicity of crystal structures, all corresponding subgroups are finite subgroups of the Euclidean group $E(3)$. For non-orthogonal lattice struc-

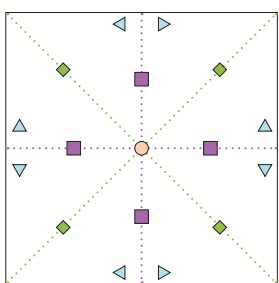

Figure 2: A 2D example of a unit cell with p4mm symmetry. Applying a group element to this set permutes "atoms" of the same type (shape and color). The symmetry divides the unit cell (black box) into four Wyckoff position: the center, horizontal and vertical coordinate axes, diagonal axes, and general position (denoted by white space).

tures, the space group acts on fractional coordinates as elements of the special affine group $SA(3)$, rather than $E(3)$, fig. 5 demonstrates this in 2D. In simpler terms, we apply the group action after mapping every lattice to the cube using $L^{-1}$.

**Wyckoff Positions.** Intuitively, Wyckoff positions of a space group $G$ indicate regions with specific symmetry properties. Atoms in general position occupy the least symmetric position in the crystal, appear most frequently in the unit cell, and enjoy the fewest restrictions on their coordinates. Meanwhile, atoms in one of the several special positions occupy a region of higher symmetry, appear less frequently in the unit cell, and are restricted to lie in low-dimensional affine subspaces. More formally, a Wyckoff position $w$, is defined by a set of $|w| = m$ affine projections $\{(V_i, \tau_i)\}_{i=1}^m$ onto the corresponding affine subspace, with $V_i \in \mathbb{R}^{3 \times 3}$ and $\tau_i \in \mathbb{R}^3$. We denote projection of $f \in [0, 1)^3$ onto each of these $m$ affine subspaces by,

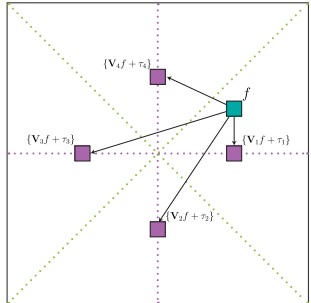

Figure 3: Projection of a coordinate $f$ using Wyckoff position $w$.

$$w(f) := \{V_i f + \tau_i - \lfloor V_i f + \tau_i \rfloor\}_{i=1}^m . \qquad (1)$$

If $w$ is a Wyckoff postion of a space group $G$, for every $y \in w(f)$ we have the property $w(f) = G \cdot y$ where $G \cdot y := \{g \cdot y \mid g \in G\}$ denotes the *orbit* of $y$ under $G$. This is visualized in fig. 3; it illustrates how $f$ is mapped to an orbit induced by $G$ through $w$. Each affine transformation $(V_i, \tau_i)$ identifies with a subgroup $G' \leqslant G$ that fixes points on its corresponding image. Namely, $y_i = V_i f + \tau_i - \lfloor V_i f + \tau_i \rfloor$ for some $f \in [0, 1)^3$ if and only if $G' = \{g \in G \mid g \cdot y_i = y_i\}$. That means that $G' = G_{y_i}$, the *site-symmetry* (stabilizer) group of $y_i$. We define the crystal $c$ (with fractional coordinates $F$) to be *W-constructable* with respect to $\mathcal{W} = (w_1, \ldots, w_k)$ if there exist $k$ points $\{x_i \in [0, 1)^3\}_{i=1}^k$ in the unit cell such that $F = \bigcup_{i=1}^k w_i(x_i)$ up to a permutation.

**Flow Matching (FM)** is a generative modeling framework that transform samples from a simple base distribution $p_0$ into a complex target distribution $p_1$ using a time-dependent diffeomorphic map, called a flow, $\psi : [0, 1] \times \mathcal{X} \to \mathcal{X}$. This flow is defined through the differential equation:

$$\frac{\mathrm{d}}{\mathrm{d}t}\psi_t(x) = v_t(\psi_t(x)), \quad \psi_0(x) = x \tag{2}$$

where $v_t : [0, 1] \times \mathcal{X} \to \mathcal{X}$ is a vector field governs the evolution of the $\psi_t$. The flow induces a time-dependent probability density path $p_t : [0, 1] \times \mathcal{X} \to \mathbb{R}$ starting at $p_0$ and ending with $p_1$. FM trains a parametric approximation $u_t$ of the true vector field $v_t$ (Lipman et al., 2024) by solving a regression objective. A conditional flow $\psi(\cdot \,|\, y) : [0, 1] \times \mathcal{X} \to \mathcal{X}$ transports the entire base distribution to a single target point $y \in \mathcal{X}$ and is governed by vector field $v_t(\cdot \,|\, y)$, which is easy to compute, unlike $v_t(\cdot)$. Lipman et al. (2022) demonstrated that optimizing $u_t$ to match $v_t(\cdot \,|\, y)$ with regression leads to the same optimum as matching the marginal vector field $v_t$.

## 3 METHOD

This section presents our proposed model, SGFM, a flow matching-based generative approach designed to sample crystal structures conditioned on specified space groups and Wyckoff positions. We start by formalizing the problem and defining the target distribution we aim to sample from in section 3.1. Then, section 3.2 outlines sufficient conditions for a flow model to sample from this distribution. Section 3.3 provides a detailed overview of the model, including its key components, the noise prior and vector field. Finally, section 3.4 presents the training details of SGFM.

### 3.1 PROBLEM DEFINITION

Given a finite set of crystals $\{c_1, \ldots, c_m\}$, each associated with a space group and Wyckoff positions $(G_i, \mathcal{W}_i)$ and drawn from an unknown target distribution $q$, our goal is to design a generative model that samples crystals $c \sim p_1$ such that $p_1 \approx q$. To incorporate the structural information encoded by $G$ and $\mathcal{W}$, we factorize the distribution as $p_1(c) = p_1(c \mid G, \mathcal{W})q(G, \mathcal{W})$, enabling us to model the crystal distribution conditionally on $G$ and $\mathcal{W}$. While $G$ and $\mathcal{W}$ are jointly sampled from empirical distribution, the proposed generative model focuses on sampling crystals from the conditional distribution $c \sim p_1(\cdot \mid G, \mathcal{W})$, which satisfies the following properties:

- $p(\cdot \,|\, G, \mathcal{W})$ is a *G-symmetric* distribution, meaning that $p(c \,|\, G, \mathcal{W}) > 0$ if $c$ is $G$-symmetric. The distribution assigns positive probability exclusively to crystals for which $G$ is their corresponding space group.
- $p(\cdot \,|\, G, \mathcal{W})$ is a *W-constructable* distribution, meaning that $p(c \,|\, G, \mathcal{W}) > 0$ if $c$ is $\mathcal{W}$-constructable. $p$ only supports crystals with fractional coordinates constructible by $\mathcal{W}$.

While the definitions provide a full description of the required distribution, they can be further simplified. The following lemma shows that, given $G$ and $\mathcal{W}$, where $\mathcal{W}$ denotes the Wyckoff positions associated with the space group $G$, a $\mathcal{W}$-constructable distribution implies a $G$-symmetric distribution.

**Lemma 3.1.** *Let $G \leqslant E(3)$ be a space group, $\mathcal{W}$ a corresponding set of Wyckoff positions and $\boldsymbol{F} \in [0, 1)^{n \times 3}$ a $\mathcal{W}$-constructable set of points in the unit cell, then $\boldsymbol{F}$ is $G$-symmetric.*

The proof (appendix A.3) relies on the fact that the action of a group element on an orbit defines a bijection.

### 3.2 THEORETICAL ANALYSIS

We now present the theoretical concepts underpinning the development of our generative model. The following theorem establishes the conditions under which a flow-based model can sample from a $G$-invariant distribution. This result has been proven in prior work by Köhler et al. (2020); Rezende et al. (2019); Song et al. (2023). For completeness, we state the theorem here and provide a concise version of its proof in appendix A.1, as it serves as a foundational component for our theoretical analysis.

---

**Algorithm 1** Sample $\boldsymbol{F}_0 \sim p_0(\cdot|G, \mathcal{W})$

---

1: **Input:** $\mathcal{W} = \{w_1, \ldots, w_k\}$ s.t. $w_i$ is a Wyckoff position of the space group $G$.
2: **Output:** $\boldsymbol{F}_0 \in [0, 1)^{n \times 3}$ s.t. $n = \sum_{i=1}^{k} |w_i|$.
3: set $\boldsymbol{F}_0 = [\,]$
4: **for** $i = 1$ to $k$ **do**
5:     Sample $x \sim \mathcal{U}[0, 1]^3$
6:     $\boldsymbol{F}_0 = \text{Concatenate}([\boldsymbol{F}_0, w_i(x)])$
7: **end for**
8: return $\boldsymbol{F}_0$

---

**Theorem 3.2.** *The probability path $p_t(x)$ defined by a flow generated by a $G$-equivariant vector field $u_t$ from a $G$-invariant prior $p$ is $G$-invariant for all $t \in [0, 1]$.*

The proof of this theorem consists of two parts. First, we demonstrate that the flow $\psi_t(x)$, is $G$-equivariant. Second, we show $p_t(x)$ is $G$-invariant. Building on this proof, we derive the conditions under which a flow-based model can sample from a distribution that is $\mathcal{W}$-constructable. Achieving this requires extending the standard framework with two modifications:

1. Introducing a noise prior that is itself $\mathcal{W}$-constructable.

2. Ensuring that the vector field model $u_t$ is equivariant with respect to the space group $G$ and the permutation group $S_n$. To do so we extend the vector field (and corresponding flow) equivariance to the group product $G \times S_n$.

**Theorem 3.3.** *The probability path $p_t(x)$ defined by a flow generated by a $G \times S_n$ equivariant vector field $u_t$ from a $\mathcal{W}$-constructable prior $p$ is $\mathcal{W}$-constructable for all $t \in [0, 1]$.*

This theorem follows directly from the lemma below (proof in appendix A.2) , which establishes that if the initial point $c_0$ of a $G \times S_n$ equivariant flow is a $G$-symmetric crystal structure, any point along the flow will be mutually $G$-symmetric with $c_0$. Based on the mutual $G$-symmetry, we demonstrate that the flow preserves the site-symmetry structure of $c_0$. This implies that if $c_0$ is $\mathcal{W}$-constructable, then $\psi_t(c_0)$ is also $\mathcal{W}$-constructable. Figure 1 (b) visualizes the core idea behind the theorem, illustrating how the equivariant vector field constrains atoms to move solely within the image of their Wyckoff position.

**Lemma 3.4.** *Let $\psi_t$ be a $G \times S_n$ equivariant flow and $c \in \mathcal{C}$ be $G$-symmetric and $\mathcal{W}$-constructable, then $\psi_t(c)$ is $G$-symmetric and $\mathcal{W}$-constructable.*

### 3.3 SGFM

In this section, we introduce the key components of SGFM, with emphasis on the prior model and the learned vector field architecture. We explain how the previously outlined conditions are concretely implemented. The main focus lies in the interaction between the method and the crystal's fractional coordinates, due to their strong dependence on the space group and Wyckoff positions.

**Noise Prior.** According to theorem 3.3, for the generated distribution to satisfy the conditions outlined in section 3.1, the noise prior must also satisfy the same constraints. Algorithm 1 presents a noise prior sampling pseudocode (with 2D visualizations in fig. 1 (a)) that generates initial fractional coordinates compliant with these requirements. The sampling procedure for the lattice parameters and atom types is described in section 3.4 for the reader's convenience. The algorithm iterates over the set of Wyckoff positions and samples orbits induced by $G$ by projecting random points from the unit cell. It follows directly from the algorithm's construction that $\boldsymbol{F}_0$ is $\mathcal{W}$-constructable (and also $G$-symmetric).

**Space Group Conditional Vector Field.** As noted previously, $u_t$ must be $G \times S_n$ equivariant in order for the flow to be $G$-symmetric and $\mathcal{W}$-constructable. The challenge lies in using a single $u_t$ model across crystals with varying space groups. Since some space groups (with non-orthogonal lattice structure) act on the fractional coordinates with special affine structured transformations, (see fig. 5), using an $E(3) \times S_n$-equivariant model is non-trivial. In addition, modeling $G$-equivariance using a super group restricts the expressiveness of the model. To address these limitations, we adopt

Group Averaging (GA) (Yarotsky, 2022), a symmetrization operator that projects a backbone model onto the space of $G$-equivariant functions. It is defined as:

$$\hat{u}(c \,|\, G) = \frac{1}{|G|} \sum_{g \in G} g \cdot u(g^{-1} \cdot c). \tag{3}$$

Applying GA to enforce space group symmetry addresses the previously mentioned limitations. Specifically, if the backbone $u$ is $S_n$ equivariant , then $\hat{u}(\cdot \,|\, G)$ is $G \times S_n$ equivariant (Puny et al., 2021). Additionally, GA is not limited to subgroups of $E(3)$ and support all finite groups. Importantly, Puny et al. (2021) showed that symmetrization preserves the expressive power of the original model. However, a drawback of GA is its computational burden: directly applying eq. (3) increases the number of evaluations of $u$ by a factor of $|G|$, which can be as large as 192 for 3D space groups (the average space group size in the MP-20 dataset is $\sim 45$). To mitigate this computationally intensive formulation, we leverage the fact that the inputs to $\hat{u}(\cdot \,|\, G)$ are $G$-symmetric crystals, allowing us to derive an efficient and equivalent formulation of GA specific to this case.

**Lemma 3.5.** *Let $c \in C$ be a crystal, $G$ its space group, and $u$ an $S_n$ equivariant vector field. Then, eq.* (3) *can be equivalently rewritten as follows:*

$$\hat{u}(c \,|\, G) = \frac{1}{|G|} \sum_{g \in G} g \cdot \sigma_{g^{-1}|c} \cdot u(c) \tag{4}$$

*Where $\sigma_{g^{-1}|c} \in S_n$ satisfy the equation $\sigma_{g^{-1}|c} \cdot c = g^{-1} \cdot c$.*

The formulation presented in eq. (4) requires only a single evaluation of $u$, which dramatically improves the model efficiency. Figure 1 (c) compares between eq. (3) and eq. (4) and visualize the efficiency gain. Furthermore, computing $\sigma_{g^{-1}|c}$ is computationally efficient, since we can decompose the problem according to the orbits of $c$, determined by $\mathcal{W}$ (appendix A.3). At inference time, these permutations only need

Table 1: Training and generation time comparison of different vector field models.

| Model | Training | | Generation |
|---|---|---|---|
| | Batch size | Time (s) | Time (s) |
| SGFM | 64 | 28.2 | 17.81 |
| Non-Equivariant | 64 | 26.3 | 16.39 |
| GA | 1 | 600 | 14.2 |

to be computed once for $c_0$, since theorem 3.3 guarantees that the flow preserves $G$-symmetry structure. During training, permutations are computed only once during preprocessing for every data point. Table 1 compares the training and generation runtimes between SGFM, a non-equivariant variant (no symmetrization), and standard GA highlighting the efficiency gains of our GA implementation compared to the standard GA, and demonstrating that its computational cost is comparable to using a backbone without symmetrization. Further details of this comparison are in appendix H.

## 3.4 TRAINING SGFM

This section provides an overview of the SGFM training process. Let $c_1 \in \mathcal{C}$ be a crystal from the training set with a corresponding space group $G$ and Wyckoff positions $\mathcal{W}$ . We will denote $c_0 \sim p_0(\cdot|G, \mathcal{W}, c_1)$ a sample from the conditional noise prior, $c_t = \psi_t(c_0 \,|\, c_1)$ the conditional flow where $c_t = (k_t, \boldsymbol{F}_t, \boldsymbol{A}_t)$, $v_t(c_t|c_1) = (v_t^k(c_t|c_1), v_t^{\boldsymbol{F}}(c_t|c_1), v_t^{\boldsymbol{A}}(c_t|c_1))$ is the conditional vector field and $\hat{u}_t(c_t|G) = (\hat{u}_t^k(c_t|G), \hat{u}_t^{\boldsymbol{F}}(c_t|G), \hat{u}_t^{\boldsymbol{A}}(c_t|G))$ is the prediction of the $G \times S_n$ equivariant vector field parametric model.

**Lattice Parameters.** As noted in section 2, we represent lattice parameters using the group-conditioned form from (Jiao et al., 2024), where $k \in \mathbb{R}^6$ encodes the basis coefficients of a 3D symmetric matrix constrained to $G$-specific subspaces. To sample $k_0 \in \mathbb{R}^6$, we first draw coefficients $k' \sim \mathcal{N}(0, \boldsymbol{I})$ and apply a group condition mask: $k_0 = k' \odot m(G)$, where $m(G) \in \{0, 1\}^6$ is a group-dependent binary mask that zeros out the irrelevant coefficients. For hexagonal lattices we set the first entry of $k_0$ to be $\frac{-\log(3)}{4}$, as described in table 5 $k_t$ is computed as a linear interpolation of $k_0$ and $k_1$, $k_t = (1 - t)k_0 + tk_1$ and the corresponding component of the conditional vector field is $v_t^k(c_t|c_1) = k_1 - k_0$. Since the group action does not directly act on the lattice parameterization we need to apply the $m(G)$ on $\hat{u}_t^k(c_t|G)$, both in training and after each generation step. The lattice optimization objective is:

$$\mathcal{L}^k(\theta) = \mathbb{E}_{t, q(c_1), p_0(c_0|G)} \left\| \hat{u}_t^k(c_t|G) \odot m(G) - (k_1 - k_0) \right\|_2^2 \tag{5}$$

**Atom Types.** For the DNG task, which involves predicting atom types, we follow the modeling approach introduced in Miller et al. (2024), where atom types are represented using a $\{-1, 1\}$

binary format instead of standard one-hot encoding. Specifically, $A_1 \in \{0,1\}^{n \times h}$ is converted into its binary representation $\tilde{A}_1 \in \{-1,1\}^{n \times \lceil \log_2 h \rceil}$. To ensure $G$-symmetry in the initial sample $c_0$, atom types must be consistent within each orbit. Accordingly, we sample initial Gaussian noise $\mathcal{N}(0,1)^{\lceil \log_2 h \rceil}$ per orbit and broadcast it to all atoms within that orbit to sample $\tilde{A}_0 \in \{-1,1\}^{n \times \lceil \log_2 h \rceil}$. We define $\tilde{A}_t = (1-t)\tilde{A}_0 + t\tilde{A}_1$ and $v_t^A(c_t|c_1) = \tilde{A}_1 - \tilde{A}_0$. The atom types optimization objective is:

$$\mathcal{L}^A(\theta) = \mathbb{E}_{t,q(c_1),p_0(c_0|G)} \left\| \hat{u}_t^A(c_t|G) - (\tilde{A}_1 - \tilde{A}_0) \right\|_2^2 \tag{6}$$

Our GA formulation (eq. (4)) ensures that $\hat{u}_t^A(c_t \mid G)$ is $G$-invariant, meaning the atom type vector field is consistent across orbits, as required. Detailed explanation about this property can be found in appendix G.3. During inference, we apply the sign function to convert the continuous atom type predictions into their binary representation.

**Fractional Coordinates.** Algorithm 1 describes a general procedure for sampling fractional coordinates that are both $G$-symmetric and $\mathcal{W}$-constructable. To ensure $G$-symmetry of the conditional flow, the initial coordinates $\boldsymbol{F}_0 \sim p_0(\cdot,|,G,\mathcal{W})$ must be $G$-symmetric with $\boldsymbol{F}_1$. This requires that the order of elements and operators in $\mathcal{W}$ match that used to generate $\boldsymbol{F}_1$, which we precompute during preprocessing using the PyXtal library (Fredericks et al., 2021). We adopt the flat torus geometry of the unit cell, following the approach proposed by Miller et al. (2024), and define the conditional flow over the fractional coordinates,

$$\psi_t(\boldsymbol{F}_0|\boldsymbol{F}_1) = \boldsymbol{F}_0 + t \cdot \log_{\boldsymbol{F}_0}(\boldsymbol{F}_1), \tag{7}$$

$$\log_{\boldsymbol{F}_0}(\boldsymbol{F}_1) = \frac{1}{2\pi} \operatorname{atan2}([\sin(\boldsymbol{F}_1 - \boldsymbol{F}_0), \cos(\boldsymbol{F}_1 - \boldsymbol{F}_0)]). \tag{8}$$

Where $\log_{(\cdot)}(\cdot)$ is the element-wise logarithmic map over the flat tori. In appendix B we demonstrate: (1) the conditional vector field $\log_{\boldsymbol{F}_0}(\boldsymbol{F}_1)$ is $G$-equivariant but with respect to a different representation of $G$. Let $g \in G$ then $\log_{g \cdot \boldsymbol{F}_0}(g \cdot \boldsymbol{F}_1) = g^\star \log_{\boldsymbol{F}_0}(\boldsymbol{F}_1)$ where $g^\star$ is defined by the homomorphism $(\boldsymbol{R}, \tau) \mapsto \boldsymbol{R}$; (2) $\psi_t(\boldsymbol{F}_0|\boldsymbol{F}_1)$ is mutually $G$-symmetric with $\boldsymbol{F}_0$ and $\boldsymbol{F}_1$, hence $G$-symmetric and $\mathcal{W}$-constructable. The fractional coordinates optimization objective is:

$$\mathcal{L}^F(\theta) = \mathbb{E}_{t,q(c_1),p_0(c_0|G,\mathcal{W})} \left\| \hat{u}_t^{\boldsymbol{F}}(c_t|G) - \log_{\boldsymbol{F}_0}(\boldsymbol{F}_1) \right\|_2^2 \tag{9}$$

Combining all components and hyperparameters $\lambda_k, \lambda_{\boldsymbol{F}}, \lambda_{\boldsymbol{A}} \in \mathbb{R}^+$ we obtain our training objective:

$$\mathcal{L}^{\text{SGFM}}(\theta) = \lambda_k \mathcal{L}^k(\theta) + \lambda_{\boldsymbol{F}} \mathcal{L}^{\boldsymbol{F}}(\theta) + \lambda_{\boldsymbol{A}} \mathcal{L}^{\boldsymbol{A}}(\theta). \tag{10}$$

## 4 EXPERIMENTS

The experiments can be divided into two sections: *Crystal Structure Prediction* (CSP) implies predicting the fractional coordinates and lattice parameters given atom types and number of atoms in the unit cell. In practical CSP tasks in materials science the atom types and the number of atoms in the unit cell is unknown, but our CSP benchmark remains a useful unit test nevertheless. In this task, knowledge of the correct Wyckoff positions provides a significant advantage. We therefore differentiate DiffCSP++ and SGFM from the other models in this experiment, but provide evaluations of models without this knowledge for completeness. Since our contribution is primarily about effecitvely conditioning on space groups and Wyckoff positions, the most important comparison is between models that have access to the ground truth wyckoff positions. This unit test evaluation removes the ambiguity of inaccurate Wyckoff positions for matching to ground truth structures. We also provide results using a method where the Wyckoff positions are inferred using a heurstic method (Kusaba et al., 2022). We test on five datasets and perform ablation studies to assess our method.

In the second task *De Novo Generation* (DNG), we generate the atom types along with the fractional coordinates and lattice parameters to accurately simulate a computational materials discovery campaign. We evaluate models based upon the number of thermodynamically stable, unique, and novel (S.U.N.) structures they generate. We consider two instantiations of our model: SGFM when we use the empirical distribution (training data) to provide Wyckoff positions and the setting where a model proposed Wyckoff positions (Kazeev et al., 2025). We train on the experimental MP-20 dataset.

**Datasets.** We evaluate our method on five datasets: *MP*-20 (Jain et al., 2013), with 45,231 diverse crystals from the Materials Project; *MPTS*-52, a time-ordered variant with $40,476$ crystals featuring larger unit cells; and *Alex-MP*-20, a large-scale set of $607,684$ crystals combining MP-20 and Alexandria data (Schmidt et al., 2022a;b). We also assess CSP on two unit-test style datasets: *Perov*-5 (Castelli et al., 2012), with $18,928$ perovskites sharing a common structure but varying atom types, and *Carbon*-24 (Pickard, 2020), containing $10,153$ carbon crystals with diverse structures.

**Baselines.** We compared SGFM to several state-of-the-art baselines. Methods that do not incorporate space group information in their generation process include *CDVAE* (Xie et al., 2021), *ADiT* (Joshi et al., 2025), *FlowMM* (Miller et al., 2024), *FlowLLM*, *MatterGen* (Zeni et al., 2023), DiffCSP (Jiao et al., 2023), (Sriram et al., 2024), and *OMatG* (Hoellmer et al., 2025). In contrast, *SymmCD* (Levy et al., 2025), *DiffCSP++* (Jiao et al., 2024), *WyFormer* (Kazeev et al., 2025), and *SGEquiDiff* (Chang et al., 2025) explicitly incorporate space group information. Additional details on each baseline are provided in appendix C.

**Model Details.** To model $\hat{u}_t$, we adopt the architecture used in Miller et al. (2024), which utilizes *EGNN* (Satorras et al., 2022) to handle fractional coordinates. The model applies sinusoidal embeddings to the fractional coordinates, ensuring invariance to lattice translations in addition to the space group equivariance. A description of the architecture and the hyperparameters used in each experiment are provided in appendix G. For improved sampling quality, we apply inference anti-annealing (Yim et al., 2023; Bose et al., 2023) that adjusts the prediction velocity during generation.

Table 2: Crystal Structure Prediction. MR denotes match rate. *uniform* implies a uniform base distribution, *GA* denotes group average. Best results are in bold within groupings regarding access to explicit Wyckoff positions. Access categories are {None, Predict with CSPML, Ground truth}. (we will fill in missing RMSE values, there is a logistical issue in getting them at the moment)

| Model | Wyckoff Positions | MP-20 MR (%) ↑ | RMSE ↓ | MPTS-52 MR (%) ↑ | RMSE ↓ | Perov-5 MR (%) ↑ | RMSE ↓ | Carbon-24 MR (%) ↑ | RMSE ↓ | Alex-MP-20 MR (%) ↑ | RMSE ↓ |
|---|---|---|---|---|---|---|---|---|---|---|---|
| CDVAE | | 33.90 | .1045 | 5.34 | .2106 | 45.31 | .1138 | 17.09 | .2969 | - | - |
| DiffCSP | | 51.49 | 0.0631 | 12.19 | 0.1786 | 52.02 | 0.0760 | 17.54 | **0.2759** | - | - |
| FlowMM | None | 61.39 | **.0566** | 17.54 | **.1726** | 53.15 | **.0992** | **23.47** | .4122 | - | - |
| OMatG | | **69.83** | .0741 | **27.38** | .1970 | **83.06** | .3753 | - | - | 72.50 | .1260 |
| SGFM (uniform) | | 64.49 | - | - | - | - | - | - | - | - | - |
| CSPML | Predict | 70.51 | - | 36.98 | - | 51.84 | - | - | - | - | - |
| DiffCSP++ | with | **70.58** | .0272 | **37.17** | .0676 | 52.17 | .0841 | - | - | - | - |
| SGFM | CSPML | 70.13 | - | 35.09 | - | **54.10** | - | - | - | - | - |
| DiffCSP++ | Ground | 80.27 | .0295 | 46.29 | .0896 | 98.44 | .0430 | - | - | 83.18 | **.0188** |
| SGFM (no GA) | truth | 68.16 | - | - | - | - | - | - | - | - | - |
| SGFM | | **82.74** | **.0288** | **51.79** | **.0827** | **98.57** | **.0188** | **55.02** | **.0952** | **84.40** | .0198 |

## CRYSTAL STRUCTURE PREDICTION

The generative task in CSP requires sampling from the conditional target distribution $c \sim q(\cdot|\mathbf{A})$, where $\mathbf{A}$ denotes a predefined atom type composition. This conditioning implies that during both training and generation $\mathbf{A}_t = \mathbf{A}$ for all $t \in [0,1]$, effectively ignoring the loss term $\mathcal{L}^A$ and the atom type component $\hat{u}_t^A(c_t|G)$. For evaluation, a crystal structure is generated for each entry in the test set and then compared against the corresponding ground truth structure using pymatgen `StructureMatcher` (Ong et al., 2013) with same threshold values as in Jiao et al. (2024). We report two metrics: the match rate (MR), defined as the fraction of generated structures that successfully match their ground truth counterparts, and the RMSE, averaged over all matched pairs. We conduct CSP on all datasets and with multiple ablations that we explain below. Results in table 2.

**Ground truth Wyckoff positions (with efficient Group Averaging)** In this section we quantify CSP performance among models that access the ground truth Wyckoff positions. We determine how well each algorithm can utilize this useful, but in-practice unobserved, information. We compare our model SGFM, with and without *(no GA)* the efficient group average, to the only baseline method that performs this task: DiffCSP++. In this setting, SGFM outperforms others across all datasets achieving state-of-the-art results on every metric except Alex-MP-20 RMSE. Comparing CSP accuracy as a function of generation steps (fig. 4), we observe that SGFM reaches near-optimal accuracy within just 50–100 steps. DiffCSP++ converges more slowly, requiring up to 1000 steps to approach its best performance—while still showing a notable gap in match rate compared to SGFM.

This difference is likely due to SGFM's flow matching formulation. In this same plot, we include a version of SGFM without anti-annealing to assess the effects of anti-annealing on performance.

We include SGFM (no GA) without the group average as an ablation study. The model remains conditional on ground truth Wyckoff positions because the prior depends on them; however, the vector field is not conditional nor equivariant with respect to $G$. Performance degrades by a significant factor.

**Predicting Wyckoff positions with CSPML** Now that we know conditioning on the ground truth Wyckoff positions can produce such strong results, how can apply our method with the Wyckoff positions are unknown? We utilize *CSPML* (Kusaba et al., 2022), a metric learning-based model that, given an atom type composition, retrieves a similar composition from a template set—along with its associated space group and Wyckoff positions. We then use those Wyckoff positions as input to SGFM and DiffCSP++ to perform the benchmark. The differences between CSPML, which also includes a method for predicting atomic coordinates; DiffCSP++; and SGFM are relatively minor according to table 2. We take this result as evidence that correctly guessing the Wyckoff positions is an extremely important step in accurately performing the CSP unit test. Methods that directly predict atomic coordinates outperform the CSPML-conditional results, implying that there is room for innovation in predicting Wyckoff positions.

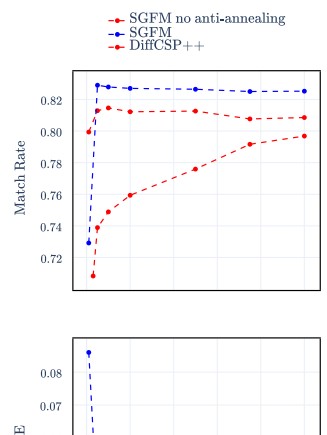
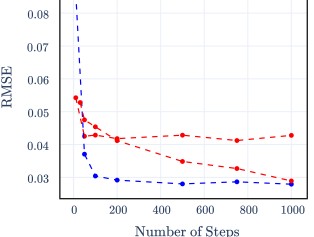

Figure 4: MR ($\uparrow$) and RMSE ($\downarrow$) versus integration steps on MP-20.

**Results without Wyckoff positions** We completely ablate the space group conditional aspects of our model and apply it without the group average and without the Wyckoff position conditional prior, replacing it with a uniform prior on atomic coordinates. SGFM (uniform) becomes a test of the architecture itself. Performance is competitive with OMatG on MP-20. Considering all the ablations and comparing to models that have access to ground truth Wyckoff positions, the gap between SGFM (uniform) and SGFM (no GA) is smaller than between SGFM (no GA) and SGFM. This comparison quantifies the value of a Wyckoff conditional prior compared to the group average.

Table 3: DNG evaluation. Models were trained on the MP-20 dataset. NFE refers to the number of generation steps per sample. We evaluated all the S.U.N. metrics, except those marked with *. Methods above the dividing line do not explicitly use Wyckoff positions; methods below use them.

| Model | NFE | Validity (%) $\uparrow$ | | Property $\downarrow$ | | | Stable (%) $\uparrow$ S.U.N (%) $\uparrow$ | | Stable (%) $\uparrow$ S.U.N (%) $\uparrow$ | |
| | | Structural | Composition | $d_\rho$ | $d_{elem}$ | $d_{cn}$ | $E_{hull} < 100$ meV/Atom | | $E_{hull} < 0$ meV/Atom | |
| --- | --- | --- | --- | --- | --- | --- | --- | --- | --- | --- |
| CDVAE | 5000 | **100.00** | 86.70 | 0.688 | 0.278 | - | - | - | - | - |
| ADiT | 500 | 99.74 | **92.14** | - | - | - | **72.0** | 27.4 | 13.0 | 4.6 |
| FlowMM | 500 | 96.86 | 83.24 | 0.075 | 0.079 | 0.443 | 31.2 | 19.7 | 4.6 | 2.3 |
| FlowLLM | 250 | 99.81 | 89.05 | 0.660 | 0.090 | - | 67.9 | 21.9 | 14.2 | 3.6 |
| MatterGen | 1000 | 100.00 | 82.60 | 0.206 | 0.242 | - | - | 24.3 | - | - |
| OMatG | 680 | 95.05 | 82.84 | **0.060** | **0.017** | 0.165 | 44.4 | 23.7 | 6.6 | 2.2 |
| SymmCD | 1000 | 90.34 | 85.81 | 0.230 | 0.400 | - | - | - | - | - |
| DiffCSP++ (empirical) | 1000 | 99.94 | 85.12 | 0.235 | 0.374 | - | 31.4 | 21.1 | 7.2 | 4.0 |
| DiffCSP++ (Wyformer) | 1000 | 99.66 | 80.34 | 0.670 | 0.098 | - | - | - | - | 3.8* |
| SGEquiDiff | 1000 | 99.25 | 86.16 | 0.193 | 0.209 | - | - | 25.8* | - | - |
| SGFM (empirical) | 500 | 99.87 | 86.81 | 0.075 | 0.181 | **0.076** | 64.1 | **30.3** | **14.6** | **6.9** |
| SGFM (Wyformer) | 500 | 99.87 | 84.76 | 0.237 | 0.233 | - | 48.4 | 22.6 | 10.6 | **4.7** |

## DE NOVO GENERATION

We evaluate the ability of our generative model to discover thermodynamically stable, unique, and novel crystals; identify the validity of the generated samples; and investigate divergences between property distributions. Novel structures do *not* appear in the training or validation splits of MP-20. Results and baselines are shown in table 3. SGFM conditioned on empirical Wyckoff positions produces SOTA results overall. SGFM using Wyformer is SOTA at the tighter stability threshold.

These are the evaluation metrics: *Validity %* defines two different heuristics that realistic crystals should satisfy. *Structural validity* implies that the pairwise atomic distances of a crystal's atoms are all greater than 0.5Å. *Compositional validity* implies that a crystal has a neutral charge according to so-called SMACT (Davies et al., 2019) rules. The *properties* that we consider for computing divergences include $\rho$ the atomic density defined by number of atoms divided by unit cell volume, elem (airity) the number of unique elements in a crystal, and cn (coordination number) or the the number of bonds per atom on average. We report the Wasserstein divergence between the test set and a structurally and compositionally valid subset of 1000 generated samples. Finally, we compute the *thermodynamic stability*, *novelty*, and *uniqueness* of generated crystals. Thermodynamic stability implies a structure is at or near a local minima in composition space. This requires a short explanation which can be read in appendix D. We then compute the uniqueness and novelty of each stable crystal (*S.U.N.*) against other generations and the train and validation set, respectively using `StructureMatcher` (Ong et al., 2013) with default settings.

Table 4: Generation times for various batch sizes.

| Model | Generation Time (s/batch) | | |
|---|---|---|---|
| | 64 | 256 | 500 |
| DiffCSP++ | 66 | 268 | 490 |
| SGEquiDiff | 19 | 75 | 154 |
| SGFM | 22 | 92 | 175 |

We trained SGFM on the MP-20 dataset, including an atom type prediction module, and generated structures from each of our configurations for evaluation. The configurations include using Wyckoff positions taken from the train set, denoted *empirical*, and from the output of *Wyformer* (Kazeev et al., 2025), with an eponymous denotation. All systems were evaluated with $10,000$ samples, except DiffCSP++ (empirical) that uses only $1,000$ samples. DiffCSP++ (Wyformer) (Kazeev et al., 2025) and SGEquiDiff (Chang et al., 2025) are reported results with slightly different density functional theory settings and only 100 relaxations, respectively. To further assess the generation efficiency of SGFM relative to other models, we measured generation times (across multiple batch sizes) on a single NVIDIA L40S GPU. The results are reported in table 4. Despite requiring more parameters (16.2M compared to 12.2M for DiffCSP++ and 5.5M for SGEquiDiff) and operating on the full unit cell rather than the asymmetric unit (which is more memory-efficient), SGFM achieves comparable efficiency to SGEquiDiff and outperforms DiffCSP++.

## 5 RELATED WORK

There is a growing body of literature about generative models for inorganic crystals. We focus here on works with similar inductive biases, namely explicit utilization of Wyckoff positions. We first consider works that generate atomic coordinates. Cao et al. (2024) created an autoregressive model that generates crystals sequentially in Wyckoff position's lexicographic order. Jiao et al. (2024); Levy et al. (2025) produced diffusion models that both represent crystals within the asymmetric unit, a memory-efficient formulation that contains just one representative per orbit. Neither of these methods utilize space group equivariance and both require projection steps to keep atomic coordinates within the target Wyckoff positions. A concurrently developed diffusion model by Chang et al. (2025) also utilizes the asymmetric unit; however, it implements space group equivariance via group averaging and does not require projection. Working in the asymmetric unit does not allow for our efficient reformulation in eq. (4) since representing the crystal structure through the asymmetric unit does not yield a $G$-symmetric representation. As written, Levy et al. (2025) do not address the crystal structure prediction problem. There are also a class of models that generate coarse-grained Wyckoff positions alone, ignoring explicit atomic coordinates. (Zhu et al., 2024; Kazeev et al., 2025; Kelvinius et al., 2025) both take this approach, inspired by regression methods (Goodall & Lee, 2020; Goodall et al., 2022). These models synergize with ours and can generate Wyckoff positions for SGFM to use in DNG in section 4. Innovations in these methods, if conditional on atom types, could replace CSPML for CSP. Further discussion of other relevant work is left for appendix E.

## 6 CONCLUSIONS

In this work, we introduced SGFM, a FM based generative model for crystal structures, conditioned on space group and Wyckoff positions. By design, SGFM produces crystals that satisfy symmetry constraints, relying on sufficient conditions we formulated over the noise prior and vector field. We also implemented an efficient group averaging method, enabling the incorporation of space group equivariance into the vector field model with minimal overhead. Evaluated on both CSP and DNG

tasks, SGFM achieved state-of-the-art performance. Future directions include extending the model to an unconditional generation setting, where space group and Wyckoff positions are also generated rather than specified.

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

# A  PROOFS

## A.1  PROOF OF THEOREM 3.2

*Proof.* The proof has two main parts. First, we will show that the flow $\psi_t$ defined by the $G$-equivariant vector field $u_t$ is $G$-equivariant. Then, we will use this property to demonstrate that the resulting probability path $p_t$ is $G$-invariant. As a reminder, the flow $\psi : [0,1] \times \mathcal{X} \rightarrow \mathcal{X}$ is defined by the following ODE:

$$\frac{d}{dt}\psi_t(x) = u_t(\psi_t(x)) \tag{11}$$

$$\psi_0(x) = x \tag{12}$$

To demonstrate that $\psi_t$ is equivariant, we will show that two functions, $\varphi_t(x) := \psi_t(g \cdot x)$ and $\phi_t(x) = g \cdot \psi_t(x)$ (for arbitrary $g \in G$) satisfy the same ODE with identical initial conditions.

$$\frac{d}{dt}\varphi_t(x) = \frac{d}{dt}\psi_t(g \cdot x) = u_t(\psi_t(g \cdot x)) = u_t(\varphi_t(x))$$
$$\varphi_0(x) = \psi_0(g \cdot x) = g \cdot x$$

$$\frac{d}{dt}\phi_t(x) = \frac{d}{dt}g \cdot \psi_t(x) = g \cdot \frac{d}{dt}\psi_t(x) = g \cdot u_t(\psi_t(x)) = u_t(g \cdot \psi_t(x)) = u_t(\phi_t(x))$$
$$\phi_0(x) = g \cdot \psi_0(x) = g \cdot x$$

Where the forth equality uses the $G$-equivariance of $u_t$. We can therefore conclude $\psi_t(g \cdot x) = g \cdot \psi_t(x)$ for every $x \in \mathcal{X}, g \in G$ and $t \in [0,1]$, which prove that $\psi_t$ is $G$-equivariant.

It remains to show that $p_t$ defines an invariant probability path.

$$p_t(g \cdot x) = p_0(\psi_t^{-1}(g \cdot x)) \det\left[\frac{\partial \psi_t^{-1}}{\partial x}(g \cdot x)\right]$$
$$= p_0(\psi_t^{-1}(x)) \det\left[\frac{\partial \psi_t^{-1}}{\partial x}(g \cdot x)\right]$$
$$= p_0(\psi_t^{-1}(x)) \det\left[g \cdot \frac{\partial \psi_t^{-1}}{\partial x}(x) \cdot g^{-1}\right]$$
$$= p_0(\psi_t^{-1}(x)) \det\left[\frac{\partial \psi_t^{-1}}{\partial x}(x)\right] = p_t(x)$$

The second equality follows from the $G$-equivariance of $\psi_t^{-1}$ and the $G$-invariance of $p_0$. The third equality is a consequence of the definition of the Jacobian matrix for equivariant functions, and the final equality relies on standard properties of the determinant.

### A.2 Proof of Theorem 3.4

*Proof.* The first part of the proof, which involves showing that $\psi_t$ is $G$-symmetric, is straightforward and follows directly from the equivariance properties of $\psi_t$. Since $\psi_t$ is $G \times S_n$ (and the group actions commute) it trivial to see that it is equivariant to each of the groups separately. Let the $g \in G$ then:

$$g \cdot \psi_t(c) = \psi_t(g \cdot c) = \psi_t(\sigma \cdot c) = \sigma \cdot \psi_t(c)$$

Where the first equality follows from the $G$-equivariance of $\psi_t$, the second holds because $c$ is $G$-symmetric, and the final equality follows from the $S_n$-equivariance of $\psi_t$. From the above equation, we also conclude that $c$ and $\psi_t(c)$ are mutually $G$-symmetric. Now, let $g' \in G_{f_i}$, meaning the $g'$ belongs to the site-symmetry group of $f_i$, the $i^{\text{th}}$ fractional coordinate of $c$. Since $g' \in G$ there exist a permutation $\sigma' \in S_n$ s.t $g' \cdot c = \sigma' \cdot c$. Moreover, because $g' \in G_{f_i}$, the permutation must fix the index $i$, $\sigma'(i) = i$. From the previous part of the proof, we know that $g' \cdot \psi_t(c) = \sigma' \cdot \psi_t(c)$ and since $\sigma'(i) = i$, it follows that $g' \in G_{f'_i}$, where $f'_i$ is the $i^{\text{th}}$ fractional coordinate of $\psi_t(c)$. Therefore, $\psi_t(c)$ retains the same site-symmetry structure as $c$, and is thus also $\mathcal{W}$-constructable.

$\square$

### A.3 Proof of Theorem 3.1

let $g \in G$, our goal is to show that there exists a permutation $\sigma \in S_n$ such that $g \cdot \boldsymbol{F}_0 = \sigma \cdot \boldsymbol{F}_0$. Since $F_0$ is $\mathcal{W}$-constructable it can be written as a union of orbits under the action of $G$. Focusing on a single orbit generated by $w_i$, and denote it as $\boldsymbol{F}_0^{w_i}$ we can observe that $g \cdot \boldsymbol{F}_0^{w_i} = \sigma' \cdot \boldsymbol{F}_0^{w_i}$ for some $\sigma' \in S_{|w_i|}$. This holds because the action of a group element on an orbit is a bijection. Repeating this process for each orbit contained in $\boldsymbol{F}_0$ yields a construction for the permutation $\sigma$. $\square$

### A.4 PROOF OF THEOREM 3.5

*Proof.*

$$\hat{u}_t(c|G) = \sum_{g \in G} g \cdot u_t(g^{-1} \cdot c) = \sum_{g \in G} g \cdot u_t(\sigma_{g^{-1}|c} \cdot c) = \sum_{g \in G} g \cdot \sigma_{g^{-1}|c} \cdot u_t(c)$$

the second equation comes from the $G$-symmetry of $c$ and the last comes from the $S_n$ equivariance of $u_t$. □

## B CONDITIONAL FLOW ON FRACTIONAL COORDINATES

**Lemma B.1.** *Let $G$ be a space group, the flat tori logarithmic map $\log_{\boldsymbol{F}_0}(\boldsymbol{F}_1)$ is $G \times S_n$ equivariant.*

*Proof.* let $g \in G$ such that $g = (\boldsymbol{R}, \tau)$. Since the orthogonal components of $G$ maps the crystal to itself, it preserve the lattice structure. combining with the following lemma (which is expressed with respect to a single point) we get that logarithmic maps is equivariant with respect to the space group and that the representation the acts on the output domain includes only the orthogonal part without the translation. The $S_n$ equivariance is trivial for an element wise function. □

**Lemma B.2.** *Let $g \in G$ such that $g = (\boldsymbol{R}, \tau)$. if $\boldsymbol{R}$ maps $\mathbb{Z}^3$ to itself $\boldsymbol{R} \log_x(y) = \log_{g \cdot x}(g \cdot y)$.*

*Proof.* Let $\log_x(y) = v$ that means that exist $z \in \mathbb{Z}^3$ s.t $v = y - x + z$ where $v \in [-\frac{1}{2}, \frac{1}{2})^3$. Now lets assume $\log_{g \cdot x}(g \cdot y) = v'$, that means that there exist $z' \in \mathbb{Z}^3$ s.t $v' = g \cdot y - g \cdot x + z'$. plugging in $g = (\boldsymbol{R}, \tau)$ results in $v' = \boldsymbol{R}(y - x) + z'$. combining both equations we get that $v' = \boldsymbol{R}v + z''$ (because $\boldsymbol{R}z \in \mathbb{Z}^3$). Since $v' \in [-\frac{1}{2}, \frac{1}{2})^3$ and $\|v\| = \|\boldsymbol{R}v\|$ we conclude that $v' = \boldsymbol{R}v$.

□

**Lemma B.3.** *The conditional flow $\psi_t(\boldsymbol{F}_0|\boldsymbol{F}_1)$ is $G$-symmetric and $\mathcal{W}$-constructable.*

*Proof.*

$$\begin{aligned}
g \cdot \psi_t(\boldsymbol{F}_0|\boldsymbol{F}_1) &= (\boldsymbol{F}_0 + t \log_{\boldsymbol{F}_0}(\boldsymbol{F}_1))R^T + \boldsymbol{1}_n \tau^T \\
&= \boldsymbol{F}_0 R^T + \boldsymbol{1}_n \tau^T + t \log_{\boldsymbol{F}_0}(\boldsymbol{F}_1) R^T \\
&= g \cdot \boldsymbol{F}_0 + t \log_{g \cdot \boldsymbol{F}_0}(g \cdot \boldsymbol{F}_1) \\
&= \sigma \cdot \boldsymbol{F}_0 + t \log_{\sigma \cdot \boldsymbol{F}_0}(\sigma \cdot \boldsymbol{F}_1) \\
&= \sigma \cdot \psi_t(\boldsymbol{F}_0|\boldsymbol{F}_1)
\end{aligned}$$

The fact that $\psi_t(\boldsymbol{F}_0|\boldsymbol{F}_1)$ is $\mathcal{W}$-constructable follows directly from the proof in appendix A.2 and the fact the $\psi_t(\boldsymbol{F}_0|\boldsymbol{F}_1)$ is mutually $G$-symmetric with $\boldsymbol{F}_0$ and $\boldsymbol{F}_1$. □

## C LATTICE REPRESENTATION

The lattice matrix $\boldsymbol{L} \in \mathbb{R}^{3 \times 3}$ characterizes the geometry of the unit cell. When $\boldsymbol{L}$ corresponds to a physically valid lattice, i.e., it has positive volume, it is invertible and can be decomposed to the product $\boldsymbol{L} = \boldsymbol{Q} \exp(\boldsymbol{S})$ where $\boldsymbol{Q} \in \mathbb{R}^{3 \times 3}$ is an orthogonal matrix and $\boldsymbol{S} \in \mathbb{R}^{3 \times 3}$ is a symmetric matrix. Representing the lattice parameters via $\boldsymbol{S}$ enjoys the benefits of orthogonal invariance (any orthogonal transformation is added to $\boldsymbol{Q}$), which makes this representation invariant to any space group operations. Jiao et al. (2024) suggested representing $\boldsymbol{S}$ using the coefficients of the following basis -

$$\boldsymbol{B}_1 = \begin{pmatrix} 0 & 1 & 0 \\ 1 & 0 & 0 \\ 0 & 0 & 0 \end{pmatrix}, \boldsymbol{B}_2 = \begin{pmatrix} 0 & 0 & 1 \\ 0 & 0 & 0 \\ 1 & 0 & 0 \end{pmatrix}, \boldsymbol{B}_3 = \begin{pmatrix} 0 & 0 & 0 \\ 0 & 0 & 1 \\ 0 & 1 & 0 \end{pmatrix},$$

$$\boldsymbol{B}_4 = \begin{pmatrix} 1 & 0 & 0 \\ 0 & -1 & 0 \\ 0 & 0 & 0 \end{pmatrix}, \boldsymbol{B}_5 = \begin{pmatrix} 1 & 0 & 0 \\ 0 & 1 & 0 \\ 0 & 0 & -2 \end{pmatrix}, \boldsymbol{B}_6 = \begin{pmatrix} 1 & 0 & 0 \\ 0 & 1 & 0 \\ 0 & 0 & 1 \end{pmatrix}.$$

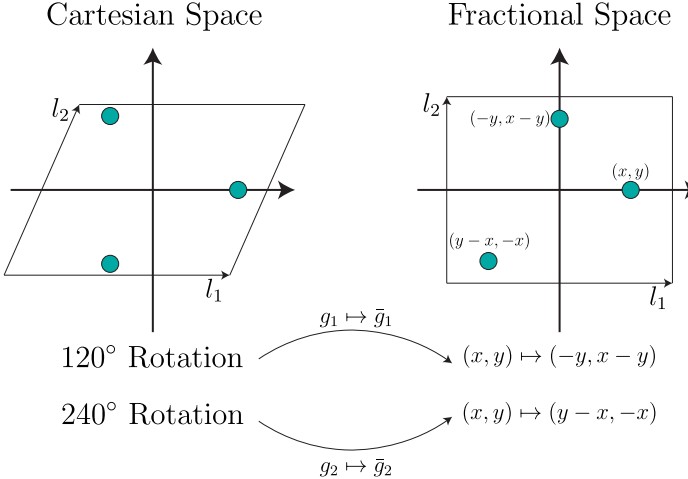

Figure 5: 2D visualization of the deformed geometry induced by moving to fractional coordinates in a non-orthogonal lattices basis. This example demonstrates how a 3-fold rotational space group becomes a set of special affine transformations when acting on fractional coordinates and, specifically, the mapping of each rotation element to a corresponding special affine transformation. In addition, the figure illustrates how a space group induced orbit transforms under this change of basis. This new representation makes it difficult to use models that rely on Euclidean geometry.

This basis enables clustering of the crystallographic space groups based on the basis coefficients used to represent $S$. Table 5 summarizes the lattice and coefficient constraints for each crystal family type.

Table 5: Relationship between the lattice shape and the constraint of the symmetric bases.

| Crystal Family | Space Group No. | Lattice Shape | Constraint of Symmetric Bases |
|---|---|---|---|
| Triclinic | $1 \sim 2$ | No Constraint | No Constraint |
| Monoclinic | $3 \sim 15$ | $\alpha = \gamma = 90°$ | $k_1 = k_3 = 0$ |
| Orthorhombic | $16 \sim 74$ | $\alpha = \beta = \gamma = 90°$ | $k_1 = k_2 = k_3 = 0$ |
| Tetragonal | $75 \sim 142$ | $\alpha = \beta = \gamma = 90°$ $a = b$ | $k_1 = k_2 = k_3 = 0$ $k_4 = 0$ |
| Hexagonal | $143 \sim 194$ | $\alpha = \beta = 90°, \gamma = 120°$ $a = b$ | $k_2 = k_3 = 0, k_1 = -log(3)/4$ $k_4 = 0$ |
| Cubic | $195 \sim 230$ | $\alpha = \beta = \gamma = 90°$ $a = b = c$ | $k_1 = k_2 = k_3 = 0$ $k_4 = k_5 = 0$ |

## D DENSITY FUNCTIONAL THEORY

Crystals exist in competition for stability between alternatives with the same composition. If one plots energy against composition, the lowest energy structures form a *convex hull*. We say a crystal is thermodynamically stable if it is near or below this convex hull. Since we do not know all structures, there is epistemic uncertainty in this characterization. The difference between the energy of a crystal and this convex hull is denoted $E_{\text{hull}}$. We report $E_{\text{hull}} < 100$ meV/atom and $E_{\text{hull}} < 0$ meV/atom rates for stability metrics. These values are computed by prerelaxation with a machine learning interatomic potential (Barroso-Luque et al., 2024) followed by relaxation and energy evaluation using density functional theory.

For the stability metrics, we applied the Vienna ab initio simulation package (VASP) (Kresse & Furthmüller, 1996) to compute relaxed geometries and ground state energies at a temperature of 0 K and pressure of 0 atm. We used the default settings from the Materials Project (Jain et al., 2013) known as the `MPRelaxSet` with the PBE functional (Perdew et al., 1996) and Hubbard U corrections. These correspond with the settings that our prerelaxation network OMat24 (Barroso-Luque et al., 2024) was trained on, so prerelaxation should reduce DFT energy, up to fitting error.

We did *not* make any guesses about oxidation states! This deviates from the Materials Project which does make those guesses. For this reason, our energy above hull calculations for structures that need to consider oxidation state are slightly high, implying that we might be under-predicting stability. This applies to any stability result we calculated. We expect it to also be a negligible effect.

The results from DiffCSP++ WyFormer in table 3 were computed by Kazeev et al. (2025) and differ slightly from ours. Specifically, they run a multiple relaxations to avoid errors that come from using a poor initial guess before relaxation. Since we prerelax with OMat24 we expect that double relaxation is unnecessary. Consult their work for further details, but we believe the differences are negligible for this purpose.

## E  RELATED WORK

As a continuation from section 5, we discuss other related work. We still limit the focus to the most-relevant parts of this large body of literature.

Our method resembles non-deep learning based methods that propose structures using Wyckoff positions as inductive bias (Glass et al., 2006; Pickard & Needs, 2011) and refine the atomic positions using density functional theory. This field is known as high-throughput screening of inorganic crystals and it is responsible for generating several important datasets of stable materials (Saal et al., 2013; Kirklin et al., 2015; Wang et al., 2021; Schmidt et al., 2022a;b). Recently, those searches have been sped up by machine learning interatomic potentials that closely approximate density functional theory (Merchant et al., 2023).

Now we take a step further away conceptually to discuss methods that are tangentially related to ours. *Crystal-GFN* (Mila AI4Science et al., 2023) is a G-flow network that uses the space group, but does not consider Wyckoff positions. *Mat2Seq* (Yan et al., 2025) proposes a one-dimensional sequence representation of crystal structures that embeds space group information. *GMTNet* (Yan et al., 2024) enforces space group invariance in a crystal property prediction model. Several other works generate crystals without considering multiple types of atom (Wirnsberger et al., 2022), or molecule (Köhler et al., 2023). Additionally, there is a large and growing cannon of generative models for materials that do not have general space group equivariance (Xie et al., 2021; Yang et al., 2023; Zeni et al., 2023; Miller et al., 2024; Sriram et al., 2024; Lin et al., 2024; Joshi et al., 2025; Hoellmer et al., 2025).

## F  BASELINES

We provide additional context on the core approach behind each baseline we compared against: *CDVAE*, integrates a diffusion model with a variational autoencoder for crystal structure generation; *ADiT*, which use latent-based diffusion model and train on additional information from the QM9 (Wu et al., 2018) dataset; *FlowMM*, an application of Riemannian Flow Matching (Chen & Lipman, 2024) that incorporates non-trivial geometries in the crystal representation space; *FlowLLM*, combines FlowMM with a Large Language model that uses as base distribution generator. *OMatG*, leverages Stochastic Interpolants (Albergo et al., 2023) for material generation; *SymmCD*, operates on the asymmetric unit and incorporates Wyckoff positions as part of the generative process; *DiffCSP++*, a diffusion-based model that conditions on space groups and projects each denoising step through Wyckoff position transformations; *WyFormer*, which employs an autoregressive model to generate Wyckoff positions (conditioned on space group) and subsequently uses DiffCSP++ model for full structure generation to predict the structure; and finally, *SGEquiDiff* a diffusion based model that enforce space group equivariance while working on the asymmetric unit.

# G MODEL DETAILS

## G.1 ARCHITECTURE

In this section, we present a comprehensive overview of our vector field model $\hat{u}_t(\cdot \,|\, G)$, along with the hyperparameters employed during training and generation across all experiments. The model takes as input a crystal $c = (k, \boldsymbol{F}, \boldsymbol{A})$, where $f^i \in \mathbb{R}^3$ denotes the $i^{\text{th}}$ fractional coordinate in $\boldsymbol{F}$, and $a^i \in \{0,1\}^h$ represents the $i^{\text{th}}$ atom type indicator vector in $\boldsymbol{A}$. The forward computation of $s$ layers model $\hat{u}(c, |, G)$ is defined by the following set of equations:

$$a^i_{embed} = \phi^a(a^i)$$
$$t_{embed} = \text{SinusoidalTimeEmbedding}(t)$$
$$h^i_{(0)} = \phi^{\text{embed}}([a^i_{embed}, t_{embed}])$$
$$m^{ij}_{(l)} = \phi^{\text{edge}}_{(l)}\left(\left[h^i_{(l-1)}, h^j_{(l-1)}, k, \text{SinusoidalEmbedding}(\log_{f^i}(f^j)), \frac{L^T L \log_{f^i}(f^j)}{\|L^T L \log_{f^i}(f^j)\|}\right]\right)$$
$$m^i_{(l)} = \frac{1}{n}\sum_{j=1}^{n} m^{ij}_{(l)}$$
$$h^i_{(l)} = \phi^{\text{node}}_{(l)}\left(\left[h^i_{(l-1)}, m^i_{(l)}\right]\right)$$
$$u^k_t(c_t) = \phi^k\left(\frac{1}{n}\sum_{j=1}^{n} h^i_{(s)}\right)$$
$$(u^{\boldsymbol{F}}_t(c_t))^i = \phi^{\boldsymbol{F}}(h^i_{(s)})$$
$$(u^{\boldsymbol{A}}_t(c_t))^i = \phi^{\boldsymbol{A}}(h^i_{(s)})$$
$$u_t(c_t) = (u^k_t(c_t), u^{\boldsymbol{F}}_t(c_t), u^{\boldsymbol{A}}_t(c_t))$$
$$\hat{u}_t(c_t \,|\, G) = \sum_{g \in G} g \cdot \sigma_{g^{-1}|c} \cdot u_t(c_t)$$

We denote $d$ as the hidden dimension of the model, $d_t$ as the Sinusoidal Time Embedding dimension, and $d_s$ as the Sinusoidal Embedding dimension. Next, we list the learnable modules constructing the model and denote their input and output dimension as $x \to y$. $\phi^a$ is a linear layer $h \to d$, $\phi^{\text{embed}}$ is a linear layer $d + d_t \to d$ dimension $d$, $\phi^{\text{edge}}_{(l)}$ is 2-layer Multi-layer Perceptron (MLP) $2d + d_t + 9 \to d$, $\phi^{\text{node}}_{(l)}$ is a 2-layer Multi-layer Perceptron (MLP) $2d \to d$, $\phi^k$ is a linear layer $d \to 6$, $\phi^{\boldsymbol{F}}$ is a linear layer $d \to 3$ and $\phi^{\boldsymbol{A}}$ is a linear layer $d \to h$. The last equation represents the group averaging presented in eq. (4). The flat tori logarithmic map is defined by the equation:

$$\log_{f^i}(f^j) = \frac{1}{2\pi} \text{atan2}\left(\left[\sin(f^j - f^i), \cos(f^j - f^i)\right]\right) \tag{13}$$

Table 6 summarize the hyperparameters used to train our SGFM models. Note that the same configuration was applied uniformly across all datasets and tasks. The hyperparameter search was performed on MP-20 (CSP) and the resulting settings were adopted for all other experiments.

## G.2 TRAINING & GENERATION

All of our models were trained using the ADAM optimizer (Kingma & Ba, 2014) on 8 NVIDIA A100 GPUs. Table 7 outlines the training configuration for each model, including the ranges explored during hyperparameter search. We employed a cosine annealing learning rate schedule with a minimum learning rate of 0.00001. As described in section 4, we applied inference anti-annealing to enhance generation quality. This technique modifies the vector field by scaling it with a time-dependent function $s(t) = 1 + s't$, where $s' \in \mathbb{R}^+$ is a hyperparameter. We defined separate annealing parameters for each crystal component: $s'_{\boldsymbol{F}}$ and $s'_k$ (no annealing was applied to atom type prediction). For the CSP experiments, we set $s'_{\boldsymbol{F}} = 3$, $s'_k = 3$, and for DNG, we used $s'_{\boldsymbol{F}} = 5$, $s'_k = 3$. Especially for Alex-MP-20 CSP we used $s'_{\boldsymbol{F}} = 5$, $s'_k = 3$. All datasets have $60/20/20$ train/validation/test split

Table 6: Hyperparameter details for all the models reported in the paper. Hyperparameter (bottom row) search was conducted on the MP-20 dataset.

| Dataset | Number of Layers | $d$ | $d_t$ | $d_s$ | Activation | Layer Norm |
|---|---|---|---|---|---|---|
| | | | CSP | | | |
| MP-20 | 8 | 512 | 256 | 128 | SiLU | ✓ |
| MPTS-52 | 8 | 512 | 256 | 128 | SiLU | ✓ |
| Carbon-24 | 8 | 512 | 256 | 128 | SiLU | ✓ |
| Perov-5 | 8 | 512 | 256 | 128 | SiLU | ✓ |
| Alex-MP-20 | 8 | 512 | 256 | 128 | SiLU | ✓ |
| | | | DNG | | | |
| MP-20 | 8 | 512 | 256 | 128 | SiLU | ✓ |
| | | | Hyperparameter Range | | | |
| MP-20 | $\{6, 7, 8, 9\}$ | $\{128, 256, 512\}$ | $\{128, 256\}$ | $\{128, 256\}$ | - | - |

Table 7: Training hyperparameter details for all the models reported in the paper. Hyperparameter (bottom row) search was conducted per experiment.

| Dataset | Batch Size/GPU | Learning Rate | Epochs | $\lambda_F$ | $\lambda_A$ | $\lambda_k$ |
|---|---|---|---|---|---|---|
| | | | CSP | | | |
| MP-20 | 64 | 0.0005 | 5000 | 100 | - | 1 |
| MPTS-52 | 32 | 0.0005 | 5000 | 100 | - | 1 |
| Carbon-24 | 64 | 0.0005 | 8000 | 100 | - | 1 |
| Perov-5 | 256 | 0.0005 | 1000 | 100 | - | 1 |
| Alex-MP-20 | 32 | 0.0003 | 1250 | 100 | - | 1 |
| | | | DNG | | | |
| MP-20 | 64 | 0.0007 | 5000 | 100 | 1 | 1 |
| | | | Hyperparameter Range | | | |
| - | - | $\{0.0002, 0.0005, 0.0007\}$ | $\{1000, 1250, 2000, 5000, 8000\}$ | $\{1, 10, 50, 100\}$ | $\{1, 10, 50, 100\}$ | $\{1, 10, 50, 100\}$ |

except Alex-MP-20 that has $80/10/10$ split. We used the same split exact indexing split as OMatG to produce a train, test, and validation set (Hoellmer et al., 2025). OMatG derived its train/test split from MatterGen (Zeni et al., 2023) where it took 10% of the training data to make a validation set.

### G.3 ATOM TYPE PREDICTION PROPERTIES

This section explains the properties of the atom-prediction component of $\hat{u}(\cdot \mid G)$. Specifically, it shows that when the input is a symmetric crystal $c$, the predicted atom-type vector field $\hat{u}^A(c \mid G)$ assigns identical values to atoms belonging to the same orbit. A point that may not be immediately obvious is that the group actions are defined in a similar way on the output $\hat{u}(c|G) = (\hat{u}^k(c|G), \hat{u}^F(c|G), \hat{u}^A(c|G))$ as they are defined over $c$, up to a representation, as discussed in the section on fractional coordinates.

To begin with, we can show that $\hat{u}^A(c \mid G)$ is $G$–invariant. On one hand, we know that $\hat{u}^A(c \mid G) = \hat{u}^A(g \cdot c \mid G)$ because $\hat{u}$ is $G$–equivariant and the group element $g$ does not act directly on the atom-type component. Moreover, we also have $\hat{u}^A(g \cdot c \mid G) = \hat{u}^A(\sigma_{g|c} \cdot c \mid G) = \sigma_{g|c} \cdot \hat{u}^A(c \mid G)$ since $\hat{u}$ is also $S_n$–equivariant and $c$ is $G$–symmetric.

This relation, $\hat{u}^A(c \mid G) = \sigma_{g|c} \cdot \hat{u}^A(c \mid G)$, applies only to permutations corresponding to elements of the space group, which only permute atoms within the same orbit induced by the group action. Consequently, we can conclude that the atom-type vector field $\hat{u}_t^A$ identifies on elements belonging to the same orbit.

## H RUNNING TIME ABLATIONS

This experimental ablation study aims to evaluate the efficiency of SGFM by comparing its performance during both training and generation against two baseline models: (1) a non-equivariant variant where the vector field does not incorporate group symmetry, and (2) a standard GA imple-

mentation as defined in eq. (3). For each model, we measured the time required to train a single epoch on MP-20, as well as the time needed to generate a batch of 64 (with 500 generation steps). The training time was averaged over 10 epochs, while the generation time was averaged over 100 batches. Training was conducted on an NVIDIA RTX8000 using 8 GPUs, while generation was performed on a single NVIDIA A10 GPU. The results are summarized in table 1. Due to memory constraints, the standard GA model was limited to a maximal batch size of 1 per GPU. The generation results for the standard GA model were obtained from a randomly initialized model, as training such a model was intractable. The results highlight a significant efficiency gap between the SGFM implementation of GA and the standard version, while showing minimal difference compared to the non-equivariant model.

## I    LARGE LANGUAGE MODELS

We utilized of large language models (LLMs) to assist with language refinement and proofreading. No content, ideas, or analyses were produced by these tools. The usage was quite limited.

