# OpenReview forum: "Space Group Conditional Flow Matching"
_ICLR.cc/2026/Conference — Submitted to ICLR 2026_

### Official Review · Reviewer_SKAi · 2025-10-25

**Soundness:** 1
**Presentation:** 2
**Contribution:** 3
**Rating:** 2
**Confidence:** 4

**Summary:**

This paper proposes a way of obtaining a space group equivariant generative flow model for generation of crystals. The method relies on group averaging, and the authors use the special properties of space groups and show how the group averaging can be computed with a single evaluation of the neural network. They evaluate the method on crystal structure prediction and *de-novo* generation.

While I think the method seems theoretically sound (although I have a question related to the comparison with using symmetrical unit, “Questions”) and very interesting, my main concern is about the choice of the CSP task, how this is performed, and how the results are presented and discussed. More details are in “Weaknesses”, but this is the reason for my rating. However, based on my current overall interpretation of the paper, if the authors address my concerns and question, I am inclined to raise my rating and soundness score.

**Strengths:**

The proposed method seems theoretically sound and, as far as I know, novel, and I think it is a nice contribution. The results on *de novo* generation look ok.

**Weaknesses:**

I think that the CSP problem is in some sense a questionable choice of task for evaluation, as the proposed method requires knowledge about the Wyckoff positions, which essentially is part of what should be predicted in the CSP task. The weakness is not that the method requires Wyckoff positions, but that this task is used for evaluation as the method is not designed for this task. Now, the authors point out that they need access to the Wyckoff positions, which are not known, and on line 357 state that they need to predict them. However, on this line they do not mention how they make the prediction, nor reference where in the paper I can read about how they make this prediction. Continuing reading, I therefore interpret it as if they use some method for predicting the Wyckoff positions. **However**, as I understand it, **in the main results table (Table 2), they have used the ground truth Wyckoff positions**, and it is not until Line 412 they come back to the problem of predicting Wyckoff positions, and then present results when they have used a method for predicting. In that case, the (limited) results seem to indicate that their method perform roughly as the baseline methods, including the method they use for predicting the positions.

I think that having access to the ground truth Wyckoff positions is a very strong and unfair advantage compared to the other methods (except DiffCSP++ which also has that access) as once you have the Wyckoff positions, the method “only” has to find the remaining degrees of freedom. I think this is a big concern with the evaluation of the CSP task, especially since Table 4 demonstrates that the proposed method doesn’t really improve over the method used for predicting Wyckoff positions, indicating that the big improvements over other methods in Table 2 could be due to having access to parts of the answer (in the case of Wyckoff positions with 0 degrees of freedom, knowing the position should leave nothing more to predict). I therefore think the authors should first clarify that they are indeed using the ground truth in Table 2, potentially add a row in Table 2 with the results when using predicted Wyckoff positions (like in the DiffCSP++ paper), and early in section 4, CSP task, incorporate a more transparent discussion about the limitations of the method in this setting, and how they evaluate this.

I am also thinking about the interpretation of the ablation study on equivariant vs non-equivariant. Table 3 indicates that the proposed method with equivariant vector field still provides a significant gain. On the other hand, table 2 seems to indicate that the proposed method and DiffCSP++, which also relies on knowledge of Wyckoff positions, perform somewhat similar, although the proposed method performs better. However, this to me could indicate (again) that a large chunk of the performance gain comes from having access to the correct Wyckoff positions.

In the DNG task, however, I think this is not really a concern as I think the results there should depend less on having access to the correct or incorrect distributions of Wyckoff positions.

**Questions:**

Fig 1c is first referenced on line 87, and then there has been no mention of details about the method. Can you extend the caption so that the figure is easier to understand (i.e., what is $\sigma$?) when only having read until line 87?

Table 1: Even if you cannot train the GA model, can’t you still use an untrained (randomly initialized) model to see the generation speed?

Line 357: I think it is supposed to be “*prima faci****a***”? But even so, I am not sure what you mean by this line? I would have expected *a priori*

Minor comment, but I believe that Latin terms like *de novo*, *prima facia*, etc, in general should be italicized

Line 387: it says the method returns atom types, but those are the same as the input (which were conditioned on)?

Table 2 + Figure 4: I assume GCFM is a typo?!

When discussing the comparison with methods using the asymmetrical unit, I become a little bit confused what the benefits of the method are. What are the symmetries for which, e.g., SymmCD are **not** equivariant to? In other words, I would have liked to see some expansion of this discussion (appendix F is only a single line)

Line 472: A related work in the same category as WyCryst and WyckoffTransformer is [1]

[1] WyckoffDiff -- A Generative Diffusion Model for Crystal Symmetry, Filip Ekström Kelvinius, Oskar B. Andersson, Abhijith S. Parackal, Dong Qian, Rickard Armiento, Fredrik Lindsten, ICML 2025

---

> ### Author Response · Authors · 2025-11-17
> **Official Comment by Authors 1/2**
>
> We thank the reviewer for the thoughtful comments. Our response is provided below.
>
> Q: The lack of clarity in the CSP section regarding the source of Wyckoff positions.
>
> A: Thank you for this detailed and important question. We believe that by answering it (and updating our paper) the clarity of the section will be highly improved. We improved the clarity of the CSP section by adding the following details:
> * A clear explanation of how the Wyckoff positions are obtained for each part of the evaluation, included in the first paragraph of the experiments section.
> * In the caption of Table 2, we explicitly state that the space group and Wyckoff position information are taken from the ground truth.
> * In the paragraph that addresses the results we emphasize that the information from the ground truth improves performance
> * We address the gap between table 2 and 4.
>
> The purpose of CSP task in the paper is three fold:
> * How much does ground truth wyckoff position information help?
> * How well our model takes advantage of ground truth wyckoff conditional information?
> * When one needs to guess at the Wyckoff positions, what is the relative performance of each method?
>
> Table 2 answers the first two of these questions. DiffCSP++ and SGFM use ground truth wyckoff position information and get superior performance to competitors in all cases. Furthermore, our equivariant formulation, SGFM, also outperforms DiffCSP++ where both models have access to the same information in all cases. We are working on an alex mp-20 comparison experiment for further proof.  Further into the paper, the improved ability to accurately place atoms seems to help the DNG task significantly as well. Therefore, we believe Table 2 illustrates an important performance improvement from our method.
>
> Table 4 answers the last question. The main source of error in Table 4 comes from incorrectly guessing Wyckoff positions by CSPML. This randomness is why SGFM has similar performance compared to other methods. The Table 4 evaluation obscures the demonstrated superiority of our method (equivariance) compared to fair baselines that also have ground truth wyckoff positions, namely DiffCSP++. The gap between Table 2 and Table 4 motivates our interest in improving models that predict Wyckoff positions from atom types.
>
> Table 3 presents an ablation study examining the combined effects of the prior distribution and the vector field model. The top-right entry, “Wyckoff conditional & non-equivariant,” places atoms into fixed Wyckoff positions (i.e., those with zero degrees of freedom) and shows that, without an equivariant vector field model, the match rate drops substantially. Another takeaway from Table 3, however, is that providing the correct Wyckoff positions as the base distribution offers a slight improvement compared to using a standard uniform distribution that does not account for Wyckoff information. That contradicts the reviewer’s claim that a large chunk of the performance gain comes from just having access to the correct Wyckoff positions.
>
> Q: "Fig 1c is first referenced on line 87, and then there has been no mention of details about the method..."
>
> A: Very good point. Our intention with the figure was to clarify the relationship between the group average and our permutation-based group average. We added an explanation in the figure’s caption.
>
> Q: " Even if you cannot train the GA model, can’t you still use an untrained (randomly initialized) model to see the generation speed?"
>
> A: We have added the result to Table 1 along with additional text describing the measurement. The full GA model generation time (per batch) for batch size=1 is approximately the same time as the generation time of SGFM for batch size=64.
>
> Q: "Line 357: I think it is supposed to be “prima facia”?.."
>
> A: You are correct, we meant a priori.
>
> Q: "Line 387: it says the method returns atom types, but those are the same as the input  (which were conditioned on) "
>
> A: In the CSP task we use the ground truth space group, Wyckoff positions and atom types as conditioning to the model (as written in line 387). The space group and Wyckoff positions are used as mentioned in section 3.3 (Wyckoff conditional prior and space group equivariant vector field). The atom type conditioning is performed by using the gt atom types as input to the model for every timestep t and ignoring the relevant atom type prediction component and loss term (as written in line 382-385) and is common in other models for this task (such as DiffCSP++).
>
> Q: "Table 2 + Figure 4: I assume GCFM is a typo?"
>
> A: Correct, thank you!

---

> > ### Author Response · Authors · 2025-11-17
> > **Official Comment by Authors 2/2**
> >
> > Q: "When discussing the comparison with methods using the asymmetrical unit, I become a little bit confused what the benefits of the method are. What are the symmetries for which, e.g., SymmCD are not equivariant to? In other words, I would have liked to see some expansion of this discussion (appendix F is only a single line)"
> >
> > A: The related section includes a discussion of the asymmetric unit (we added some further explanation), and we have added a more detailed explanation in the following paragraphs. The asymmetric unit is a subset of the unit cell that contains a single representative from each orbit induced by the space group. While this compressed representation reduces memory requirements, it loses a key property that we rely on when working with the full conventional cell: the $G$-symmetry relation $g \cdot c = \sigma \cdot c$. This relation no longer holds because the asymmetric-unit representation contains only a single representative of the orbit. Without this property, we cannot apply Equation 4 for efficient group averaging and are instead forced to use Equation 3, which is computationally more expensive (see Table 1 for an example). Methods that merely operate on the asymmetric unit, such as DiffCSP++ and SymmCD without further constraints, are **not** space-group equivariant due to their architectural design. They impose space-group conditioning through projection operations, steps that would be unnecessary if the models were equivariant by construction.SGFM is equivariant by construction via (fast) group averaging. In principle, these models could be modified to be space-group equivariant; however, as noted in the first paragraph, this would require computing the full group averaging, which is intractable. Developing efficient group-averaging techniques that operate directly on the asymmetric unit is indeed an interesting research direction and could lead to memory-efficient models.
> >
> > Q: "Line 472: A related work in the same category as WyCryst and WyckoffTransformer is [1] WyckoffDiff"
> >
> > A: Thank you, we have included the citation!

---

> > > ### Comment · Reviewer_SKAi · 2025-11-25
> > > **Comment on rebuttal**
> > >
> > > I thank the reviewers for their answers. While I think the updated CSP section somewhat at least clarifies what is being done, I am still a little unsure about how to interpret the results (and I therefore wait to update my score). My concern is essentially that when using CSPML to obtain the positions (table **4**), all compared methods perform roughly equally, *including CSPML*, although still better than the other methods in table 2 (except for Perov-5). Then, you see a big bump in performance when adding the correct Wyckoff positions (table 2). This gave me the impression that a lot of the performance gains come from knowing the Wyckoff positions.
> > >
> > > So, looking at the three questions that you write down, I think the answer to the first is “a lot” (which is not surprising, as it is part of the task to figure out, and in the case of positions with zero degrees of freedom, you are given the correct answer directly). While I agree with your answer that your model better make use of this information, my takeaway is that it is still knowing the true Wyckoff positions that gives the biggest improvements.
> > >
> > > Related to the above, in the Perov task, I thought that these materials all have the same structure? Is that correct? Because here you also see almost perfect MR when using ground truth Wyckoff positions, and maybe there is an explanation here.
> > >
> > > I think the method is interesting and sound, and there is a need for it considering methods like WyCryst, Wyformer, and WyckoffDiff that aim at solving the task of finding Wyckoff positions, but lacks a method to obtain the remaining degrees of freedom (which is where your method can come in). My concern is that CSP is given a lot of attention when your method (and DiffCSP++) is not really suited for it. In my opinion, it would work better as an ablation, after the DNG task and only comparing with DiffCSP++ and potentially CSPML to answer the questions that you have listed. Or alternatively, make it more explicit in the text that this is what you intend to show with these experiments, not setting a new SOTA on this task (because I agree with akat that this is not possible to claim when using ground truth Wyckoff positions).
> > >
> > > Follow-up question about table 3: do I interpret correctly that the Wyckoff prior means using the correct ground truth positions (the number for equivariant match table 2, but does this also hold for non-equivariant)? Also, could you elaborate how the non-equivariant vector field works (just so I don’t misunderstand what it means).
> > >
> > > Some other remarks based on other reviews:
> > > * As I understand your response, I would disagree that using ground Wyckoff positions is “non-rigorous” in the same way that the CSP task itself is “non-rigorous”. My criticism (and I interpret reviewer akat the same) is that in the CSP benchmark, whether it is a realistic materials science task or not, using the ground truth Wyckoff positions is a clear advantage for you and DiffCSP++ as effectively, **part of the task in the benchmark is to actually figure out the Wyckoff positions**, but other methods are doing this implicitly. As mentioned, using the ground truth also guarantees that the atoms with zero degrees of freedom are correct. In other words, the criticism is not that it is unrealistic in the real world to have access to the true Wyckoff positions, but that it is part of the task in the benchmark to find them.
> > > * Personally, I don’t think the evaluation time is overly important in this type of application, and a couple of seconds here or there are not really important.

---

> ### Author Response · Authors · 2025-11-25
> **please wait for reformulation of the CSP section**
>
> We are in the process of reformulating the CSP section. We will provide a new version soon. Please wait for final review until we post it!

---

> ### Author Response · Authors · 2025-11-26
> **major update to CSP section**
>
> ### Additional experiment
> - We have added the Alex-MP-20 with DiffCSP++ using ground truth wyckoff positions to the CSP results table. DiffCSP++ achieves MR: 83.18 and RMSE: .0188. We also performed another training with our method using slightly different HPs (different learning rate, and anti-annealing for positions). SGFM outperforms on MR with 84.40, but slightly underperforms on RMSE with .0198.
>
> ### Additional results
> - We included DiffCSP into the CSP results table.
>
> ### Further discussion about CSP
> - We have reformulated the CSP task section entirely, emphasizing the different modalities: Ground truth wyckoff positions, CSPML guessed wyckkoff positions, and no wyckoff positions. We updated the abstract, contributions, experimental preamble, and other discussions to match. The changes to this section “Crystal Structure Prediction” are not in the revised blue because of formatting issues with wrapfig, but other changes are in blue.
> - Considering this perspective, we combined results from all CSP-related tables into a single table. The ablations are now explained in paragraphs that reference specific entries on this table. The RMSE results are computed, but not reported because there are logistical challenges. We will address the issue as soon as possible.

---

> ### Author Response · Authors · 2025-11-26
> **response to your concerns specifically**
>
> > I thank the reviewers for their answers. While I think the updated CSP section somewhat at least clarifies what is being done ...
>
> Thank you for your continued engagement with our paper and constructive critiques to improve the communication about the CSP task. As we replied above, we have completely rewritten the CSP section. We find it in better alignment with your perspective on the matter.
>
> > So, looking at the three questions that you write down, I think the answer to the first is “a lot” ...
>
> We agree that knowing the true Wyckoff positions is hugely beneficial for this task. Although perhaps pedantic, methods other than CSPML are not guessing the Wyckoff positions, but directly placing atomic coordinates into the unit cell. This is clearly specified in our new version. We would frame the point that the other methods directly predict atomic coordinates, while CSPML attempts to do some kind of “inference” on a hidden variable (the wyckoff positions). The atomic coordinates are factored through this variable in our (& DiffCSP++’s) setting. Poor inference on this variable will also reduce performance on atomic coordinates, as shown in our results.
>
> Why do it this way? We motivate our method by attempting to bring the population we sample from in DNG to be closer to the empirical distribution of symmetries. Wyformer & symmcd both have histograms showing that methods which do not explicitly utilize symmetry typically produce low-symmetry P1 unit cells. We approach the CSP task with this motivation: given we want to target the correct symmetry, how do we fit the remaining degrees of freedom?
>
> > Related to the above, in the Perov task, I thought that these materials all have the same structure?
>
> These materials do all have the same structure, but due to minor differences in floating point values some structures are not recognized as cubic using pymatgen. We utilized the same hyperparameters for symmetry detection as DiffCSP++ so we believe we have the same training, test, and validation sets here.
>
> In perov the only degree of freedom is the size of the lattice. We believe we are probably doing better on this prediction to explain our relatively small performance increase. We do not think that this particular evaluation is very important for the overall contribution of our method.
>
> > I think the method is interesting and sound, and there is a need for it considering methods like WyCryst, Wyformer, and WyckoffDiff ...
>
> We appreciate that you see the method as interesting and sound. We hope that the new text formulation better aligns with your way of thinking about the CSP task. We also emphasize that we have explicitly softened our language to specify that we are SOTA on CSP when given access to the ground truth wyckoff positions throughout the paper (and not CSP in general).
>
> We agree that a combination of Wyformer/WyCryst/etc. adapted for CSP would be an interesting idea. We also think it’s plausible that other methods could out perform CSPML on this task. However, at the moment we are leaving this for future work since it would require reformulating other methods. We make a specific note of it in the related work section.
>
> > Follow-up question about table 3: do I interpret correctly that the Wyckoff prior means using the correct ground truth positions
>
> You are correct. In the old version Table 3 meant that we used the conditional prior, i.e. given the correct wyckoff positions initially, but random coordinates within those degrees of freedom. Yes, it also held for non-equivariant.
>
> In our new table of consolidated experiments we have clarified non-equivariant by replacing the name with a (no GA) label, i.e. for SGFM (no GA) we condition the prior on G and wyckoff positions but NOT the model. We do not apply our efficient group average.
>
> > As I understand your response, I would disagree that using ground Wyckoff positions is “non-rigorous” in the same way that the CSP task itself is “non-rigorous”.
>
> Thank you for clarification, we do understand your distinction. We agree it is a clear advantage. We hope that the distinctions we have made between the different variants of this task will satisfy the concern!

---

### Official Review · Reviewer_2ExA · 2025-10-26

**Soundness:** 3
**Presentation:** 2
**Contribution:** 3
**Rating:** 4
**Confidence:** 5

**Summary:**

This paper introduces SGFM, a flow-matching-based crystal generation framework condition on a given space group and Wyckoff positions. Based on simplified group averaging specifically for space groups, the proposed framework constrains the flow matching process to preserve the given symmetries. Results on Crystal Structure Prediction (CSP) and De Novo Generation (DNG) demonstrate the performance and efficiency of SGFM.

**Strengths:**

1. The paper provides a clear and well-structured formulation of space group symmetries, specifically Wyckoff positions, and effectively applies the concept of Group Averaging (GA) to this problem.
2. Experiments showcase the superiority of SGFM over previous space-group-related methods.

**Weaknesses:**

While the proposed framework and writing are generally straightforward, several aspects are missing or unclear, including:

1. Apart from the equations of G-equivariant operations (Eq. 3-4), the design of G-invariant operation should also be explicitly provided, which is required for atom type generation.
2. The method name is inconsistant. In Table 1 and Figure 4, it is it is referred to as GCFM. Besides, in Figure 4, the name of the baseline should be DiffCSP++.
3. For the lattice flow matching, since $k_1=-\log(3)/4\neq 0$ for hexagonal crystals, directly applying a 0-1 mask in the construction of $\mathbf{k}_0$ is not quite rigorous. A more precise definition should be $\mathbf{k}_0=\mathbf{k}'\odot m(G)+\mathbf{k}_1 (1-m(G))$.
4. In line 318, $\tilde{A}_0$ appears twice in the definition of the interpolation.

**Questions:**

1. In the implementation of Eq. 4, is it computed by explicitly enumerating all $|G|$ space group elements, or simplified by averaging over each Wyckoff site only?
2. According to the theories of Group Averaging and Frame Averaging, Eq. 4 should be G-equivariant for any arbitrary $u(c)$. Hence, instead of functioning as an equivariant operator during training, Eq. 4 could also serve as a post-processing step applied only at inference. The reviewer wonders how the two non-equivariant variants in Table 3 would perform if they were additionally equipped with this post-processing step as defined in Eq. 4 during inference. This may further validate whether the proposed SGFM can also serve as a constrained plugin for pretrained unconditional flow matching models.

---

> ### Author Response · Authors · 2025-11-17
>
> We thank the reviewer for the thoughtful comments. Our response is provided below.
>
> Q: "Apart from the equations of G-equivariant operations (Eq. 3-4), the design of G-invariant operation should also be explicitly provided, which is required for atom type generation."
>
> A: That is an excellent comment. Let us break down why this claim follows directly from the definition of the group action on the crystal $c$ and from Equation (4). A point that may not be immediately obvious is that the group actions are defined in a similar way on the output $\hat{u}(c|G)=(\hat{u}^k(c|G), \hat{u}^F(c|G), \hat{u}^A(c| G))$ as they are defined over $c$ , up to a representation, as discussed in the section on fractional coordinates.
> To begin with, we can show that $\hat{u}^{\mathcal{A}}(c | G)$ is $G$–invariant. On one hand, we know that $\hat{u}^A(c| G)=\hat{u}^A(g \cdot c| G)$ because $\hat{u}$ is $G$–equivariant and the group element $g$ does not act directly on the atom-type component. Moreover, we also have $\hat{u}^A(g \cdot c| G) = \hat{u}^A(\sigma_{g|c} \cdot c| G)=  \sigma_{g|c} \cdot\hat{u}^A( c| G)$  since $\hat{u}$ is also $S_n$–equivariant and $c$ is $G$–symmetric.
> This relation, $\hat{u}^A(c| G)=\sigma_{g|c} \cdot\hat{u}^A( c| G)$, applies only to permutations corresponding to elements of the space group, which only permute atoms within the same orbit induced by the group action. Consequently, we can conclude that the atom-type vector field $\hat{u}^{\mathcal{A}}$ identifies on elements belonging to the same orbit. We add a section in the appendix explaining this.
>
> Q: "The method name is inconsistent. In Table 1 and Figure 4"
>
> A: We fixed both. Will appear in the revised version.
>
> Q: "For the lattice flow matching...applying a 0-1 mask in the construction of $k_0$ is not quite rigorous"
>
> A: Thank you for this comment. We updated the relevant part in the revised version.
>
> Q "Line 318 - $\tilde{A}_0$..."
>
> A: Fixed, thank you!
>
> Q: "In the implementation of Eq. 4, is it computed by explicitly enumerating all  space group elements, or simplified by averaging over each Wyckoff site only."
>
> A: The averaging is performed by explicitly evaluating Equation (4). Specifically, G consists of the space group elements up to lattice translations. This is typically a bigger group than the site symmetry group at a specific Wyckoff position. It is not entirely clear what is meant by “simplified by averaging over each Wyckoff site only,” and we would appreciate it if the reviewer could elaborate on this point. If the concern refers to the computational cost of the full explicit averaging in Equation (4), we note that Table 1 demonstrates that the additional computational overhead (in both time and memory) introduced by our efficient averaging formulation is minimal compared to a model without averaging.
>
> Q: "...Hence, instead of functioning as an equivariant operator during training, Eq. 4 could also serve as a post-processing step applied only at inference. The reviewer wonders how the two non-equivariant variants in Table 3 would perform if they were additionally equipped with this post-processing step as defined in Eq. 4 during inference..."
>
> A: We thank the reviewer for this insightful observation. Following the suggestion, we evaluated both non-equivariant models (from table 3), the one trained with a uniform prior and the one trained with a Wyckoff-conditioned prior, after incorporating group averaging at the inference stage. The match rate (MR) for the model with the Wyckoff-conditioned prior increased from 68.16 to 80.5, while the model trained with the uniform prior showed a decrease from 64.49 to 60.5.
> This further supports our conclusion that the main advantage of our approach arises from the combination of the Wyckoff-conditioned prior and the space group equivariant model, as described in Theorem 2. The model trained with a uniform prior lacks the inductive bias that enables it to benefit from space group equivariance, since its support is primarily over non-symmetric crystals. An interesting takeaway is that explicit group averaging during training is not strictly necessary, provided that the prior is appropriately defined. Nevertheless, given its low computational cost and the additional performance improvement it offers, incorporating group averaging during training remains beneficial.

---

> > ### Comment · Reviewer_2ExA · 2025-11-23
> >
> > Thanks to the authors for their efforts. The reviewer’s questions have been largely addressed. Regarding my original Question 1, if I understand correctly, the vector field automatically maintains W-constructability through group averaging, rather than through explicit projection onto the subspaces of specific Wyckoff positions. I have recognized the contribution of this work and have raised my score to 6.

---

> > > ### Author Response · Authors · 2025-11-24
> > > **Thank you**
> > >
> > > Thank you for reviewing our rebuttal; we sincerely appreciate your support.
> > >
> > > Your comment is correct. Due to the G-symmetry of the crystals and the G-equivariance of the vector-field model (achieved through group averaging), the resulting vector field is also G-equivariant, which in turn ensures that it remains W-constructible.

---

### Official Review · Reviewer_akat · 2025-11-01

**Soundness:** 3
**Presentation:** 3
**Contribution:** 2
**Rating:** 4
**Confidence:** 4

**Summary:**

This paper proposes a flow matching based method to generate crystals conditioned on space groups and Wyckoff positions. This is done by leveraging an efficient construction for models equivariant to space group transformations. The proposed method obtains competitive empirical performance on crystal structure prediction and unconditional generation.

**Strengths:**

- The method is sound and well motivated
- The experimental results are competitive and improve existing methods on many metrics. The detailed appendix is appreciated and should help with reproducibility and usage
- The background section of the paper is well written and accessible

**Weaknesses:**

- I find the contributions relatively incremental with respect to the recent line of work on space group conditioning of diffusion models for crystals, especially Chang et al. 2025. The main contribution is the improvement in the efficiency of the group averaging operation for equivariance, which is interesting. But I feel like the paper overstates its contributions a bit.
- I found the theoretical section overall a bit confusing. I think it is possible to reduce the amount of formalism and theorems, and would see this as an improvement.
    - In section 3.1 the paper introduces a notion of G-symmetry for distributions, which I think is confusing. The problem is that if we take the (canonical) action of G on distributions  $g \cdot P(c \mid G, W) = P(g \cdot c \mid G, W)$, the definition of a G-symmetric distribution should be a distribution satisfying  $P(g \cdot c \mid G, W) = P(c \mid G, W)$. This coincides with the notion of invariant distribution and is not the same as the definition used in the paper. I suggest that instead of talking about G-symmetric distribution, the authors say that they are interested in distributions supported on G-symmetric crystals, which is straightforward to understand.
    - It is my impression that the previous criterion (G-symmetry or support on G-symmetric crystals) is not even necessary. If a crystal is W-constructible with W compatible with G, then it is clear that the crystal is automatically G-symmetric as (see in lemma 3.4). So the analysis can be simplified by only considering W-constructibility and not G-symmetry (as defined in the paper) or support on G-symmetric crystal (as suggested above).
    - I don't think theorem 3.1 is needed. These results are already known from several previous works and can be directly used.
    - Related to the previous point, G-invariance of the distribution is discussed a few times early in the paper, but not used in the subsequent results. What is needed is G-symmetry of the support of the distribution, as said earlier, they are different.
    - I think the proof of lemma 3.3 could be simplified by using the fact that the stabilizer of the output of an equivariant function has larger stabilizer than its input (similar to Chang et al. 2025). This applies at the level of individual atomic position with stabilizer given by the site symmetry group, which should imply W-constructibility of the crystal and in turn G-symmetry.
    - I also don't see the need for theorem 3.2. As said in the text it is directly implied by lemma 3.3 and does not add to the discussion.
    - My overall feedback on the theoretical section is that it should be simplified. Lemma 3.4 implies that W-constructibility of the distribution is the necessary and sufficient criterion for generating crystals conditioned on space group. I think this fact can be mentioned in the background section. A lemma is not even necessary, this is clear from crystallography. Then you need to show that the equivariant generative model yields W-constructibility which is done through lemma 3.3 (this is similar to Chang et al.). I feel like this should be enough and much more straightforward.
- I also suggest avoiding the use of very general geometry concepts like logarithmic maps. For the torus, this is simply a modulo operation. This will be clearer to practitioners familiar with crystals and will simplify exposition. The torus is the only manifold of interest here and is not complicated enough to justify this level of abstraction.
- I would like to see a more fine-grained analysis of the impact of how the Wyckoff positions are sampled. Using Wyformer to sample the positions seems to perform worse than using empirical distributions, which is counterintuitive. Using the empirical distributions should result in a significant loss of diversity.
- The authors should include MatterGen amongst the baselines (similar to Chang et al. 2025), it is one of the state-of-the-art methods
- Evaluating efficiency based on the number of function evaluations is not best practice, since the model architectures used for these different methods are not standardized. Instead, the authors should use metrics that report actual use of compute (memory and time). Even if they are hardware dependent, they are much more useful in practice.
- Minor comments:
    - I think "constructibility" is more correct than "constructability"
    - Couple of typos
        - line 47: and, and (2)
        - line 427: realsitic -> realistic
        - line 447: "we the thermodynamic stability"

**Questions:**

- I did not understand the issue with using a standard E(n) x S_n equivariant network. Can the authors clarify that point? I also didn't understand figure 5.

---

> ### Author Response · Authors · 2025-11-17
> **Official Comment by Authors 1/2**
>
> We thank the reviewer for the thoughtful comments. Our response is provided below.
>
> Q: "I find the contributions relatively incremental"
>
> A: The work presented in [1] (contemporaneously-developed with ours) also developed a space group conditional generative model, as we do, but there are several key differences:
> * We provided a formal definition of the problem in terms of distributional constraints.
> * We derived sufficient conditions for sampling from such distributions, extending the results of [2].
> * We introduced a novel formulation for group averaging over symmetric crystals, which allows efficient operations on the entire fractional coordinate matrix rather than at the individual atom level.
>
> Q: "I found the theoretical section overall a bit confusing"
>
> A: We thank the reviewer for the detailed feedback on our theoretical section. We believe that addressing these points will help us improve the paper.
>
> Q: "…The problem is that if we take the (canonical) action of G on distributions.."
>
> A: We feel that we don’t understand the reviewer’s intention. A distribution is a scalar function $P:X\to \mathbb{R}$ therefore it is not clear how the group acts on the output space and what is the meaning of the canonical action of G on distributions.
>
> Q: "This coincides with the notion of invariant distribution and is not the same as the definition used in the paper. I suggest that instead of talking about G-symmetric distribution, the authors say that they are interested in distributions supported on G-symmetric crystals..."
>
> A: We define a G-symmetric distribution as one whose support consists of G-symmetric crystals (see definition in section 3.1), same as the reviewer suggested.
>
> Q: "It is my impression that the previous criterion (G-symmetry or support on G-symmetric crystals) is not even necessary…"
>
> A: This is a valid observation. We initially included both definitions for clarity, but based on your feedback, we will restructure the theoretical section accordingly (see revised version).
>
> Q: "don't think theorem 3.1 is needed…. G-invariance of the distribution is discussed a few times early in the paper, but not used in the subsequent results.."
>
> A: We view this work as an extension of the framework for equivariant generative modeling introduced in [2], which demonstrates that not only invariant distributional constraints ($P(g \cdot x) = P(x)$) can be maintained throughout the generative process. For this reason, we chose to explicitly reference the main theorem from [2] (and to reprove it in a simpler form) and to introduce the notion of invariant distributions as contextual background.
>
> Q: "I think the proof of lemma 3.3 could be simplified by using the fact that the stabilizer of the output of an equivariant function has larger stabilizer than its input (similar to Chang et al. 2025). This applies at the level of individual atomic position with stabilizer given by the site symmetry group, which should imply W-constructibility of the crystal and in turn G-symmetry."
>
> A: We respectfully disagree with the reviewer’s comment. Theorem 1 in [1] is established for an atom-wise function $f:\mathbb{R}^3 \to \mathbb{R}^3$, where the group action is likewise defined on $\mathbb{R}^3$. In contrast, the score model used in [1] is a space group equivariant GNN, $s_{\theta}$, which operates on the full fractional coordinate matrix $X \in \mathbb{R}^{n \times 3}$ rather than on individual coordinates. It is not obvious how Theorem 1 extends to $s_{\theta}(X)$ under a group action applied to the entire matrix.
> In our setting, both the flow and the equivariant vector-field model are defined on the full fractional coordinate matrix, and the group action is applied at the matrix level as well. Consequently, we provide a proof tailored to this setting, which requires a slightly more elaborate argument.
>
> Q: "I also don't see the need for theorem 3.2..."
>
> A: We agree that Theorem 3.2 follows directly from Lemma 3.3 (as we mention in the paper). In our view, the theorem provides a complete formulation of the extension of our framework from Köhler.
>
> Q: "I also suggest avoiding the use of very general geometry concepts like logarithmic maps. For the torus, this is simply a modulo operation"
>
> A: For a Riemannian manifold, the logarithmic map $\log_x(y)$ is defined as the tangent vector at $x$ that points along the geodesic connecting $x$ to $y$. While the modulo operator can be used to project a lifted point in $\mathbb{R}^n$ onto its corresponding point on the torus, it is not equivalent to the logarithmic map. For example, consider $x = [0.1, 0]$ and $y = [0.9, 0]$. In this case, $(y - x) \bmod 1 = [0.8, 0]$, whereas $\log_x(y) = [-0.2, 0]$, which corresponds to the shortest tangent displacement on the torus. The formulation is nevertheless simple and we can use both the general and specific names for clarity.

---

> ### Author Response · Authors · 2025-11-17
> **Official Comment by Authors 2/2**
>
> Q: "I would like to see a more fine-grained analysis of the impact of how the Wyckoff positions are sampled"
>
> A: Great comment. We conducted a detailed analysis of the Wyckoff positions sampled from Wyformer (obtained from [5]) and compared them with those in the MP20 training set. The Wyformer dataset contained 10,000 samples, whereas the training set comprised approximately 27,000. Interestingly, the Wyformer samples exhibited more unique (space group, Wyckoff position) pairs than the training set (3509 vs. 3299). Furthermore, 2,245 of the Wyformer samples correspond to pairs that are absent from the training set, underscoring the model’s capacity to explore novel symmetry configurations. This confirms the intuition that the Wyformer-generated Wyckoff samples are indeed more diverse relative to the training data.
> We can further break down the components of the S.U.N. rate to better understand the results. Although the analysis below focuses on the $E < 0$ case, the conclusions hold for both settings. As shown in Table 5, the stability rate of samples drawn from the empirical distribution exceeds that of samples generated using Wyformer-based sampling ($14.6/10.6$). The uniqueness rates are nearly identical ($13.69/9.85$), indicating that most stable generations are distinct from each other. For the novelty rate, we observe a drop of roughly $50$\% ($7.11/4.95$), suggesting that about half of the generated stable crystals are not novel, even when their Wyckoff positions are sampled from the Wyformer model. In this scenario, using the empirical distribution, which matches the data distribution on which the model was trained, results in a higher proportion of stable generations and ultimately a better S.U.N. rate.
>
> Q: "The authors should include MatterGen amongst the baselines..."
>
> A: We add MatterGen to Table 5 (DNG) using results obtained from [1]. We also intend to evaluate MatterGen on the Alex-MP20 dataset.
>
> Q: "Evaluating efficiency based on the number of function evaluations is not best practice, since the model architectures used for these different methods are not standardized. Instead, the authors should use metrics that report actual use of compute (memory and time). Even if they are hardware dependent, they are much more useful in practice."
>
> A: We agree with the reviewer that the number of function evaluations alone does not fully characterize computational complexity, although it does offer insight into generation efficiency. To provide a more complete picture of SGFN’s performance, we have added an additional runtime ablation (Table 6 in the revised version) that compares inference times for different batch sizes across the various models.
>
> Q: Typos
>
> A: Fixed, thank you!
>
> Q: "I did not understand the issue with using a standard E(n) x S_n equivariant network. Can the authors clarify that point? I also didn't understand figure 5"
>
> A:  In general, if the requirement for $u_t$ is to be $G \times S_n$–equivariant, and $G \leqslant E(n)$, then employing an $E(n) \times S_n$–equivariant architecture will naturally satisfy the $G$–equivariance condition. However, enforcing equivariance with respect to a supergroup can limit the model’s expressive capacity. We appreciate the reviewer’s helpful comment, as this clarification was not explicitly included in the original manuscript.
>
> Even without the point above,  the backbone neural architecture employed in this work (used in [1, 3, 4]) is not inherently equivariant under the group action defined in this paper (can be seen from the inference experiment we did for reviewer 2ExA); therefore, an alternative approach is needed. Frame averaging has been proposed as an efficient method for approximating group averaging over $E(n)$. However, due to the non-orthogonal geometry of the hexagonal lattice, the corresponding space-group representation is not an orthogonal matrix (with respect to the Euclidean norm), which complicates the application of standard frame averaging. Figure 5 is intended to visualize this phenomenon. Therefore we believe that our $G$ conditioned model, based on efficient implementation of group averaging, is the most suitable solution for this case.
> Please let us know if this explanation helps clarify the issue.
>
> [1] Space Group Equivariant Crystal Diffusion, Chang et al., Neurips 2025.\
> [2] Equivariant flows: Exact likelihood generative learning for symmetric densities, Kohler et al. ICML 2020.\
> [3] FlowMM: Generating Materials with Riemannian Flow Matching, Miller et al., ICML 2024.\
> [4] Space Group Constrained Crystal Generation, Jiao et al. ICLR 2024.\
> [5] Generated crystals for WyFormer, DiffCSP, DiffCSP++, WyCryst, SymmCD, CrystalFormer, MiAD & MatterGen, Kazeev et al.

---

> ### Comment · Reviewer_akat · 2025-11-24
>
> I thank the authors for their response. They have answered some of my questions and comments, but some important issues remain.
>
> - Could the statement regarding impact of invariance of the distribution be supported by experimental results? For example, the authors could use a reasonable non-invariant prior and compare with the pushforward of the invariant prior
> - Regarding the theoretical section
>     - I am not really convinced regarding the need for theorems 3.2 and 3.3, I don't see how that can't just be said in a plain sentence.
>     - My comment on the logarithmic map is that since the torus is flat and globally parameterized, invoking Riemannian manifolds/log/exponential maps seems to add unnecessary abstraction. The operation can be explicitly described as x–y–round(x–y) or something of the sort, basically subtraction modulo number of unit cells. This is in my opinion much clearer and simpler than logarithmic map.
> - The most important issue is I think the experimental evaluation, which lacks rigour in comparing the different methods and does not convince me.
>   - I am quite skeptical of the comparison of models on number of function evaluations (figure 4 and table 5). This is currently given a lot of weight in the paper which I think is misleading. The new results of table 6 show that SGEquiDiff is actually faster in generating samples. The results in SGEquiDiff also show that their approach is not even the fastest, methods like SymmCD are even faster (this should be compared against). Overall, number of function evaluations is misleading and should not be used. Table 5 should include sampling time instead like in SGEquiDiff.
>   - I had not initially noticed the fact that ground-truth Wyckoff positions were used for CSP. I agree with reviewer SKAi that this is very misleading. The performance of the method should not be compared to other methods that don't use ground-truth Wyckoff positions. If the authors want to present results on CSP (which is arguably questionable given the need for space group/Wyckoff conditioning) they should generate Wyckoff positions using CSPMP and report those results. Table 4 shows that this results in significantly worst performance. The abstract wrongly claims state-of-the-art results based on this problematic evaluation method. Also another thing I noticed, why didn't the authors include the results from DIffCSP on CSP? From what I see in that paper, DiffCSP actually performs better than the proposed method without requiring another pipeline to generate positions. Overall the presentation of these results and conclusions drawn from it are not rigorous.

---

> ### Author Response · Authors · 2025-11-25
> **Official Comment by Authors 1/2**
>
> We thank the reviewer for the feedback and are confident that the remarks will strengthen our manuscript.
>
> Q: "Could the statement regarding impact of invariance of the distribution be supported by experimental results?
>
> A: "Would you mind please explaining which statement you are referring to? We are not sure which one you mean.
>
> That being said, it seems that there might be confusion about how a distribution can be non G-invariant while also only having support only on G-symmetric crystals. The detail lies in the definition of our prior: Our prior is not G-invariant due to the canonicalized way that pymatgen outputs the `wyckoff_ops`. For each orbit within a crystal, being W-constructed via pymatgen `wyckoff_ops` produces a single choice of the |orbit| permutations of the output. Therefore the prior only has mass on one G-symmetric density. It is not G-invariant due to this canonicalization. In practice, this makes no difference because the network is S_n-equivariant and does not “see” the permutation. We will clarify this in the appendix. We want to note that this theoretical detail has no effect in practice. In principle, we can fix it by applying a random group element to the output of the prior and keep track of the corresponding permutation to resolve the issue.
>
> Q: "I am not really convinced regarding the need for theorems 3.2 and 3.3, I don't see how that can't just be said in a plain sentence."
>
> A: "We appreciate the reviewer’s goal of making the theoretical section as approachable as possible. This is also our intention and we will gladly modify the paper until it is easily understandable. It seems that we disagree on the stylistic differences about how to present these facts and perceive the theorems as useful pedagogical tools to explain our method. Since the reviewer’s concern seems primarily stylistic we hope that you will accept our modifications to increase readability and connect the paper to prior work."
>
> Q: "My comment on the logarithmic map.."
>
> A:  “We adopted this geometric formulation of the unit cell, originally introduced in flowmm, because we believe it accurately captures the cell’s structure. To improve clarity for the reader, we have added an explicit definition of the function in the main text, ensuring that practical understanding of the paper does not require prior familiarity with non-trivial geometries.”
>
> Q: “The most important issue is I think the experimental evaluation, which lacks rigour in comparing the different methods and does not convince me.”
>
> A: “We appreciate the reviewer’s focus on rigor. We emphasize to you and all reviewers that **we have rigorously evaluated our model on all standard benchmarks, all of which have precedent in literature, and we do so in the manner they have been reported in relevant works (CSP with ground truth wyckoff positions as in DiffCSP++, and DNG in many works). We report a model that is unequivocally SOTA in terms of performance on those tasks.** The reviewers have raised an important question about the framing of the CSP task, which we will discuss further below. Also other matters, especially efficiency.”
>
> Q: "I am quite skeptical of the comparison of models on number of function evaluations (figure 4 and table 5). This is currently given a lot of weight in the paper which I think is misleading..."
>
> A: "We have a few general responses:
> * We understand that you are unimpressed with comparisons in terms of number of function evaluations, but we want to emphasize that SGFM has superior results to all alternative methods in both CSP (with known wyckoff positions) and in terms of S.U.N. We do not rely on being more efficient, our model simply produces better results.
> * here may be some confusion because we introduce an efficient group averaging, but we do not claim to have the most efficient model. Merely, that our group averaging makes the computation tractable whereas without it we could not apply the group average.
> * We do believe that reporting NFE is important for reproducibility.
> * Finally, we want to emphasize that output of these models typically requires relaxation and energy computation with density functional theory (DFT). DFT is extremely slow compared with evaluating our network. The bottleneck in evaluation is not the model.

---

> > ### Author Response · Authors · 2025-11-25
> > **Official Comment by Authors 2/2**
> >
> > Now for a few specific responses:
> > * Figure 4 - The figure demonstrates the superiority of our model over DiffCSP++ (in a CSP setting with ground-truth Wyckoff positions) in terms of the accuracy–efficiency tradeoff. However, we want to note that that would put the results even further into SGFM’s favor considering that SGFM is faster per batch than DiffCSP++ at all batch sizes considered in Table 6.
> > * We are open to computing evaluation time instead of NFE in Table 5 to better identify efficient models, but it will take a bit of time since there are so many to compute.
> > * While SGEquiDiff does achieve faster generation times, it attains a lower S.U.N rate, which we consider to be the more important metric.
> > * Symmcd - While the model is indeed efficient, more so even than SGEquiDiff, its reported S.U.N rate is not comparable, which is why we chose not to include it in the comparison reported in Table 6."
> >
> > Q: "I had not initially noticed the fact that ground-truth Wyckoff positions were used for CSP. I agree with reviewer SKAi that this is very misleading..."
> >
> > A: "Thank you for pointing out this confusing matter. We hope that our response and changes to the text will satisfy your concern:
> >
> > * The benchmark we report identifies the models that use ground truth wyckoff positions with boxed names. We have made this clearer and are happy to continue to make edits to clarify this matter. Please note that this benchmark appears in DiffCSP++, so we are explicitly doing a benchmark with precedent in the literature.
> > * Good question, we will include DiffCSP results in addition to DiffCSP++. Please note that DiffCSP++ using CSPML also performs worse than DiffCSP. We viewed the results of DiffCSP as outdated considering the number of other models that outperform it, which we did include.
> > * We identified three reasons for each of the different experiments and tables in our response to reviewer SKAi. Please take a look at the response that contains the text: ‘Thank you for this detailed and important question. We believe that by answering it (and updating our paper) the clarity of the section will be highly improved. We improved the clarity of the CSP section by adding the following details:’
> >
> > General response regarding the CSP task:
> >
> > * We agree that knowing the wyckoff positions is a huge benefit for doing CSP
> > * Guessing the wyckoff positions is difficult and the main performance bottleneck in the CSPML-related methods is CSPML itself. If CSPML were improved all of the methods that used their wyckoff positions would be improved. We quantify how much better each method would be by giving ground truth wyckoff positions in Table 4 (boxed methods).
> > * There is a characterization that this CSP task with known wyckoff positions is a non-rigorous evaluation. We think that misses the point. The CSP task itself is also ‘non-rigorous’ by this definition because assigning the explicit number of atoms and their frequency in the unit cell is not known by materials scientists a priori. The CSP task is a benchmark evaluating the performance of the model to fit the distribution of positions and lattice vectors. We performed it in multiple settings (with and without wyckoff positions) and reported the results fairly and openly with labels. We have been completely rigorous and produced a method that gets SOTA performance on the CSP task (with known wyckoff position) and we argue that our performance on CSP (without wyckoff positions) is primarily hindered by incorrect guesses from CSPML. It does not reveal the value of our contributions if you merely look at the CSPML results.
> > * Finally, DNG is closer to being actually relevant for high-throughput screening of materials and clearly having a representation of the wyckoff positions is extremely helpful for that task. We argue that the improved performance in predicting positions and lattice vectors, as shown in the CSP (with known wyckoff positions) task, is likely partially responsible for our extremely strong performance on DNG. "

---

> ### Author Response · Authors · 2025-11-25
> **please wait for a reformulation of the CSP section**
>
> We are in the process of reformulating the CSP section. We will provide a new version soon. Please wait for final review until we post it!

---

> ### Author Response · Authors · 2025-11-26
> **major update to csp section**
>
> ### Additional experiment
> - We have added the Alex-MP-20 with DiffCSP++ using ground truth wyckoff positions to the CSP results table. DiffCSP++ achieves MR: 83.18 and RMSE: .0188. We also performed another training with our method using slightly different HPs (different learning rate, and anti-annealing for positions). SGFM outperforms on MR with 84.40, but slightly underperforms on RMSE with .0198.
>
> ### Additional results
> - We included DiffCSP into the CSP results table.
>
> ### Further discussion about CSP
> - We have reformulated the CSP task section entirely, emphasizing the different modalities: Ground truth wyckoff positions, CSPML guessed wyckkoff positions, and no wyckoff positions. We updated the abstract, contributions, experimental preamble, and other discussions to match. The changes to this section “Crystal Structure Prediction” are not in the revised blue because of formatting issues with wrapfig, but other changes are in blue.
> - Considering this perspective, we combined results from all CSP-related tables into a single table. The ablations are now explained in paragraphs that reference specific entries on this table. The RMSE results are computed, but not reported because there are logistical challenges. We will address the issue as soon as possible.

---

### Official Review · Reviewer_gjio · 2025-11-01

**Soundness:** 2
**Presentation:** 3
**Contribution:** 2
**Rating:** 4
**Confidence:** 4

**Summary:**

This paper introduces Space Group Conditional Flow Matching (SGFM), a generative model for crystal structures that enforces crystallographic symmetry. The method uses a symmetric noise prior and an equivariant vector field, enabled by a novel, computationally efficient formulation of Group Averaging (GA). The model achieves state-of-the-art (SOTA) results on crystal structure prediction (CSP) and de novo generation (DNG) benchmarks.

**Strengths:**

1. Methodological Rigor: SGFM intrinsically preserves symmetry via an equivariant flow, which is different from the post-hoc projection method used by models like DiffCSP++.

2. Efficient Group Averaging: The proposed efficient solves the computational bottleneck of applying equivariance to high-order space groups, making the approach practical.

3. Empirical Results: The model demonstrates SOTA performance, with clear gains on the challenging MPTS-52 dataset.

**Weaknesses:**

1. Limited Novelty & Unclear Theoretical Advantage: The work is an improved constraint enforcement method, not a new paradigm. This is acceptable, but the paper fails to clearly articulate the theoretical superiority of its equivariant flow over DiffCSP++'s projection. It states what DiffCSP++ does (projection) but not why this is theoretically flawed (e.g., error accumulation, breaking the true flow path), leaving the theoretical advantage of SGFM unclear.

2. Marginal Gains & Missing Baselines: Performance gains are moderate on some datasets (e.g., MP-20), and the key comparison to DiffCSP++ on the large-scale Alex-MP-20 dataset is missing.

3. Questionable Convergence Comparison: The claimed convergence speedup may also be a trivial consequence of adopting the Flow Matching framework or some sample tricks, which is a simple modification to implement from DiffCSP++. Related ablation experiments are needed.

4. Potential Theory-Practice Mismatch: The standard L2 regression loss does not guarantee the network learns the correct equivariant vector field. This creates a gap, as the learned field might be an incorrect path that is simply forced into symmetry by the final GA operator, rather than being the true physical path.

**Questions:**

1. Training-Inference Mismatch: Does the conventional L2 regression loss ensure the learned vector field is truly the correct equivariant field, or just a field that appears symmetric post-averaging? Is there a risk of learning an incorrect, non-physical path?

2. Missing Baseline: Can the authors provide the CSP results for DiffCSP++ on the Alex-MP-20 dataset to substantiate the SOTA claim on large-scale data?

3. Sampler Ablation: Can the authors provide an ablation study to isolate the performance gains from the sampling strategy? Can the authors provide reproducible code for this comparison?

---

> ### Author Response · Authors · 2025-11-17
>
> We thank the reviewer for the thoughtful comments. Our response is provided below.
>
> Q: "Limited Novelty & Unclear Theoretical Advantage…the paper fails to clearly articulate the theoretical superiority of its equivariant flow over DiffCSP++'s projection….why DiffCSP++ is theoretically flawed, leaving the theoretical advantage of SGFM unclear"
>
> A: We do not consider DiffCSP++ to be theoretically flawed. Similar to SGFM, it generates crystal structures by conditioning on space groups and Wyckoff positions, an approach we agree is valuable for modeling crystal generation. However, we believe that the theoretical contributions presented in our work are significant:
>
> * We formalize the problem of symmetric crystal generation through distributional constraints.
> * Based on these constraints, we derive provably sufficient conditions for their satisfaction.
> * We leverage these insights to design SGFM, a model that satisfies these constraints by construction.
> * We further exploit the symmetry property ($g \cdot c = \sigma \cdot c$ for $g$ in the space group) to develop a novel and efficient implementation of group averaging tailored to this setting.
>
> Overall, we view our theoretical contribution as providing a more principled foundation for symmetric crystal generation, with this grounding reflected in the superior empirical performance of SGFM.
>
> Q: "Marginal Gains:... Performance gains are moderate on some datasets (e.g., MP-20)"
>
> A: Overall, our results show a substantial improvement over DiffCSP++, with more than a 10% increase in match rate (MR) on the MPTS52 benchmark and consistent gains across all DNG evaluation metrics (Table 5), particularly in the SUN rate. The primary CSP metric, match rate (MR), is defined as the fraction of generated structures that successfully match their ground truth counterparts; however, this matching is evaluated only for valid crystals (i.e., those that are both compositionally and structurally valid). In the MP20 test set, only 91.1% of the crystals are valid, which serve as un upper bound for the MR (we get 82.7 compared to 80.2 of DiffCSP++).
>
> Q: "Missing Baselines - CSP evaluation of Diffcsp++ on alex - mp-20"
>
> A: We have done this calculation and find that DiffCSP++ achieves MR: 83.18 and RMSE: .0188. We also performed another training with our method using slightly different HPs (different learning rate, and anti-annealing for positions). SGFM outperforms on MR with 84.40, but slightly underperforms on RMSE with .0198.
>
> Q: "Questionable Convergence Comparison"
>
> A: We agree that part of the observed convergence speedup can be attributed to the intrinsic differences between flow matching and diffusion processes, a trend also noted in [1]. In addition, we incorporate inference anti-annealing, following the approach proposed in [1]. To isolate its effect, we included an ablation version without anti-annealing in Figure 4. The results show that even without anti-annealing, SGFM still outperforms DiffCSP++ and remains more efficient in terms of generation steps. Our intention was to compare another dimension of performance between the two models. Furthermore, in the context of the perennial debate about the advantages of flow matching versus diffusion our model is superior on both axes: faster generation and higher quality outputs.
>
> Q: "Potential Theory-Practice Mismatch"
>
> A: In our formulation, we introduce several notational conventions for the components of our vector field model. The symbol $u$ denotes our neural architecture backbone, a GNN-based network that is $S_n$-equivariant but not $G$-equivariant. This is intentional: since a single $u$ is used to support multiple space-group equivariances, we do not require $u$ itself to be $G$-equivariant.
> To incorporate conditional $G$-equivariance, we compose $u$ with a group-averaging operator, and denote the resulting model by $\hat{u}$. All L2 regression losses are computed with respect to this $G$-equivariant model $\hat{u}$. Likewise, $\hat{u}$ is the model used to predict the vector field during both training and inference.
> To reiterate, we are not interested in evaluating the output of the non-equivariant backbone $u$; only its equivariant version $\hat{u}$ is used. We hope this clarification fully addresses the reviewer’s question.
>
> [1] FlowMM: Generating Materials with Riemannian Flow Matching, Miller et al., ICML 2024.

---

> ### Author Response · Authors · 2025-11-26
> **curious about your thoughts regarding the updates to the paper**
>
> Dear reviewer gjio,
>
> As the deadline of Dec 3rd is fast approach, we are curious if our response and updates to the paper has satisfied your concerns enough to raise your score.
>
> We believe we have clarified your theoretical concerns and hope you will agree.
>
> We have also added your requested experiments (alex-mp-20, ablation of anti-annealing) with again strong performance from SGFM.
>
> Furthermore, we are curious about your thoughts on our response to so-called "theory/practice/inference mismatch." Until we get your feedback we cannot proceed since we believe we answered the issues.

---

### Author Response · Authors · 2025-12-03
**Rebuttal Summary**

As the discussion period concludes, we would like to present a comprehensive summary of our work, the key issues raised during the rebuttal, and the resulting updates to the paper.

Our work introduces Space Group Conditional Flow Matching (SGFM), a flow-matching–based approach for generating crystal structures with Wyckoff position and space-group conditioning.
The main contributions of this paper are as follows:
* SGFM attains state-of-the-art performance on de novo crystal structure generation, the central task in this domain. In particular, it achieves the highest S.U.N. rate when trained and evaluated on the MP20 dataset.
* We establish sufficient conditions for sampling from distributions that satisfy space-group and Wyckoff-position constraints, and we design a generative model (SGFM) that inherently respects these constraints without requiring any post-hoc projections to correct atom positions.
* We introduce an efficient formulation of group averaging for symmetric crystals that adds negligible computational overhead.

Key issues and our responses. We present this list concisely to highlight the primary concerns raised during the rebuttal.
* Reviewer SKAI raised concerns about the clarity of the CSP section, particularly regarding the source of the Wyckoff positions used in each experiment. In response, we fully restructured this section to ensure it is clear and unambiguous. We also extended our evaluation of the CSP task to the large-scale ALEX-MP20 dataset to further support our claims.
* As requested by reviewer akat, we added an efficiency analysis. The results confirm that SGFM’s strong performance does not come at the cost of computational efficiency.
* We substantially revised the paper to make it more accessible. This includes simplifying the theoretical section and reducing unnecessary notation.
* All remaining issues were clarifying questions from the reviewers, and we addressed them fully.

We firmly believe that the revisions made to the paper would have led to higher reviewer scores.

---

### Meta-Review · Area_Chair_pLbo · 2026-01-07

**Summary:**

This paper proposes a novel method for crystal structure generation by conditioning on their “Wyckoff positions”. Specifically, Wyckoff positions contain information of the crystal by partitioning the 3D unit cell according to the structure of the orbits of the crystals. The paper derives an objective to train equivariant denoisers conditioned on the Wyckoff positions.

**Reviewer Concerns:**

The reviewers’ main concerns are centered around (i) the rigoroucity and clarify of the proposed algorithm, and (ii) significance of the results and fairness in the comparisons with baselines. Regarding (i), the two main concerns are the follows. First, Reviewer gjio raised the concern that the L2 loss does not guarantee that the denoiser $u_t$ is equivariant. While the authors argued that only $\hat{u}_t$ needs to be equivariant, it seems that Theorem 3.5 requires $u_t$ being equivariant as well. Next, multiple reviewers commented that the method/theoretical part of the paper can be greatly simplified to improve clarity and readability. While the authors made some clarifications and revisions, I think it requires another round of review to decide if the improvements are sufficient.

On the experimental side, the main concern raised (which is raised by multiple reviewers) is that many of the baselines do not have access to the groundtruth Wyckoff positions, which are unknown in crystal structure prediction tasks. I highly recommend the authors to put more efforts in resolving this problem, by either adding a Wyckoff position prediction component in all experiments or demonstrate significant improvements compared to baselines that fully leverage this information as well.

I believe the paper has its merits and can be greatly improved following the reviewers’ comments. However, I think the paper is not ready to be published yet in its current form.

**Reviewer Scores:**

gjio: The reviewer may maintain their score as concerns such as "Potential Theory-Practice Mismatch" are not fully resolved by the rebuttal.

akat: The reviewer may maintain their score since they have major concerns regarding the presentation and clarity of the method sections.

2ExA: The reviewer will probably increase their score as their concerns are addressed.

SKAi: The main concern regarding the fairness of using groundtruth Wyckoff position is not fully resolved.

---

### Decision · Program_Chairs · 2026-01-26

Reject